# SHARE: BRIDGING SHAPE AND RAY ESTIMATION FOR POSE-FREE GENERALIZABLE GAUSSIAN SPLATTING

## ABSTRACT

While generalizable 3D Gaussian Splatting enables efficient, high-quality rendering of unseen scenes, it heavily depends on precise camera poses for accurate geometry. In real-world scenarios, obtaining accurate poses is challenging, leading to noisy pose estimates and geometric misalignments. To address this, we introduce SHARE, a novel pose-free generalizable Gaussian Splatting framework that overcomes these ambiguities. Our ray-guided multi-view fusion network consolidates multi-view features into a unified pose-aware canonical volume, bridging 3D reconstruction and ray-based pose estimation. In addition, we propose an anchor-aligned Gaussian prediction strategy for fine-grained geometry estimation within a canonical view. Extensive experiments on diverse real-world datasets show that SHARE achieves state-of-the-art performance in pose-free generalizable Gaussian splatting.

## 1 INTRODUCTION

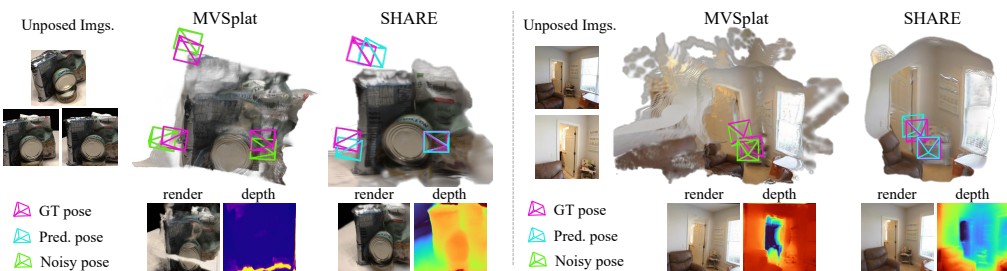

Figure 1: Given sparse-view unposed images, SHARE jointly predicts geometry, appearance, and relative camera poses. Previous generalizable 3D Gaussian splatting method (Chen et al., 2024b) is highly sensitive to camera pose and fails to reconstruct correct geometry even with a small amount of pose random noise. Meanwhile, our method demonstrates robust performance in geometry reconstruction, pose estimation, and novel view synthesis.

Recent years have witnessed unprecedented progress in Novel View Synthesis (NVS) and 3D scene reconstruction, marked by the emergence of neural implicit representations (Sitzmann et al., 2019; Park et al., 2019; Mildenhall et al., 2021) and explicit volumetric approaches such as 3D Gaussian splatting (Kerbl et al., 2023). Central to this task is the utilization of precise camera poses, which serve as a fundamental geometric prior coupling the spatial relationship between 3D space and their corresponding 2D-pixel projections across multiple views.

However, the assumption of readily available accurate camera poses is often unsatisfiable in practical scenarios. While Structure-from-Motion (SfM) (Snavely et al., 2006) techniques have long been the go-to solution for obtaining camera poses, they become increasingly unreliable as views become sparser or camera baselines widen. This challenge is particularly acute in a generalizable setting where the number of input views is often limited and test-time adaptation is not considered. A seemingly intuitive solution might be to employ learning-based camera pose prediction methods designed

for sparse view inputs. Unfortunately, even slight inaccuracies in estimated camera poses can be amplified into large positional errors in 3D space, resulting in significant geometry misalignment across input views, as shown in Figure 1.

Pose-free 3D Gaussian splatting methods have been explored to overcome this problem. Previous works (Fu et al., 2023; Fan et al., 2024) propose an iterative test-time adaptation approach that jointly rectifies pose and geometry for better alignment. GGRt (Li et al., 2024) proposes the generalizable reconstruction for video sequence inputs without pose information. Nevertheless, existing methods require additional computation for each scene or only consider sequential frames as input, compromising general applicability.

In pursuit of novel view synthesis in general scenarios without poses, we propose SHARE, a novel approach for pose-free generalizable 3D Gaussian splatting. SHARE aims to learn a holistic representation of multi-view features in a unified canonical view space. The key to our multi-view fusion process lies in embedding relative poses as spatially defined Plücker rays, which allows easy injection of pose information throughout the reconstruction pipeline. This acts as a multi-view prior across the input views in the fusion process, resolving the misalignment issues and enhancing geometric consistency across views. Building on our pose-aware fusion, we introduce anchor-based Gaussian prediction to reconstruct fine details in a unified space. We estimate pixel-aligned coarse geometry from the unified representation as *anchor points* for the local 3D space for shape consistency. From each anchor, we predict offsets to determine the splatting locations of Gaussians.

We evaluate our approach with scene level datasets such as DTU (Jensen et al., 2014) and RealEstate10K (Zhou et al., 2018) datasets. SHARE achieves robust reconstruction quality in pose-free scenarios, showing superior performance to previous pose-free generalizable reconstruction approaches (Jiang et al., 2023; Hong et al., 2024; Smith et al., 2023), and even comparable to generalizable 3DGS (Charatan et al., 2024; Chen et al., 2024b) with ground-truth poses in DTU (Jensen et al., 2014) datasets. Further analysis illustrates the effectiveness of our proposed multi-view fusion process and synergetic improvement in geometry and pose estimation. Our contribution can be summarized as below:

- We propose SHARE, a novel pose-free generalizable 3D Gaussian Splatting framework that simultaneously estimates geometry and camera pose from sparse-view unposed images.

- Our ray-guided multi-view fusion strategy effectively constructs a holistic feature for 3D shape representation, effectively mitigating the geometry misalignment while covering multi-view observations with fine details.

- SHARE shows superior performance on scene-level datasets with varying scales, including DTU (Jensen et al., 2014) and RealEstate10K (Zhou et al., 2018), outperforming existing pose-free generalizable reconstruction approaches and highlighting robustness under scene-scale datasets.

## 2 RELATED WORK

**Generalizable Novel View Synthesis with Ground-truth Camera Poses.** Generalizable novel view synthesis methods have emerged to expand the applicability of novel view synthesis (NVS) to a wider range of scenarios without test-time adaptations. Techniques based on Neural Radiance Fields (NeRF) (Mildenhall et al., 2021) utilize neural features from input views to render images by querying and accumulating features of sampled points along cast rays, leveraging known camera poses (Yu et al., 2021; Chen et al., 2021; Wang et al., 2021; Sajjadi et al., 2022). As generalizable NeRFs often face challenges in terms of rendering efficiency and processing speed due to neural network inference on densely sampled points, recent advancements have introduced 3D Gaussian Splatting for fast and efficient generalizable reconstruction (Zheng et al., 2024; Szymanowicz et al., 2024b; Charatan et al., 2024; Chen et al., 2024b; Liu et al., 2024; Wewer et al., 2024).

Generalizable 3D Gaussian splatting methods focus on predicting geometry to accurately splat Gaussians in three-dimensional space on the fly. PixelSplat (Charatan et al., 2024) learns probabilistic depth distributions in ray space, effectively addressing the local support limitation of Gaussians under rendering supervision. LatentSplat (Wewer et al., 2024) improves on this by encoding 3D variational Gaussians using a variational auto-encoder (VAE) and incorporating a discriminator to

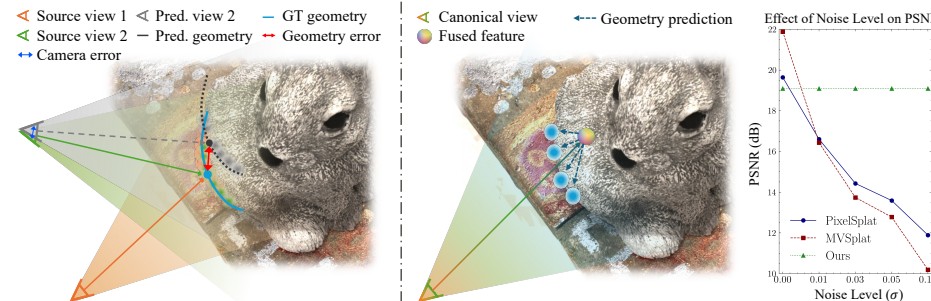

Figure 2: **Comparing SHARE with pixel-aligned generalizable 3D Gaussian Splatting (g-3DGS) in addressing geometry misalignment.** (Left) In previous g-3DGS (Charatan et al., 2024; Chen et al., 2024b), small camera pose error is amplified through depth, causing significant misalignment in 3D space. (Right) SHARE addresses this by predicting fused features in a canonical space and estimating dense Gaussians to cover multiview observations.

enhance extrapolation performance to deal with challenging large-baseline inputs where depth prediction becomes unreliable. MVSplat (Chen et al., 2024b) and MVSGaussian (Liu et al., 2024) enhance the fidelity of predicted geometry by utilizing Multi-View Stereo (MVS) to construct cost volumes via plane-sweeping of depth hypothesis. Unfortunately, these works assume to have precise camera poses, which come to be unrealistic in real-world scenarios.

**Pose-Free Generalizable Novel View Synthesis.** Several works have introduced generalizable approaches for reconstructing 3D shapes from sparse or noisy pose initializations (Hong et al., 2024; Jiang et al., 2024; Fan et al., 2023; Wang et al., 2023; Xu et al., 2024; Jiang et al., 2023). FORGE (Jiang et al., 2024) integrates camera pose estimation with radiance field prediction to achieve mutual refinement of both tasks. PF-LRM (Wang et al., 2023) implicitly leverages the power of camera estimation by parallel training of a differentiable Perspective-n-Point (PnP) solver for camera prediction. FlowCAM leverages optical flow estimation for joint flow radiance estimation, and CoPoNeRF (Hong et al., 2024) introduces joint training of pose estimation and reconstruction and correspondence matching between images. Meanwhile, LEAP (Jiang et al., 2023) liberates the need for precise pose estimation by constraining a canonical camera space, thereby constraining the pose estimation space. While these works excel in generalizable 3D reconstruction without poses, the implicit nature of their NeRF backbone complicates the joint optimization of both scene representation and camera poses (Li et al., 2024).

Gaussian-based methods (Fu et al., 2023; Fan et al., 2024) have improved optimization and efficiency for pose-free cases but continue to exhibit limitations in generalizability, requiring progressive test-time optimizations. To address this problem, concurrent pose-free generalizable 3D Gaussian Splatting (g-3DGS) methods (Smart et al., 2024; Szymanowicz et al., 2024a) utilizes pretrained models to leverage the power of geometric priors (Wang et al., 2024; Piccinelli et al., 2024). While promising, these approaches are often affected by scale ambiguities between the estimated geometry and camera poses, requiring fine-tuning with ground-truth depth supervision. GGRt (Li et al., 2024) also addresses pose-free 3D Gaussian Splatting by taking video frames as input, assuming sequential pose transformation between views. In contrast, our method is designed to handle sparse-view inputs with diverse camera baseline configurations without scale ambiguity by offering improved robustness by jointly estimating cameras as rays and 3D Gaussian splats in a unified canonical space.

## 3 OVERVIEW

Our approach addresses the critical challenge of geometric misalignment when extending 3D Gaussian splatting to unposed settings. This misalignment arises from difficulties in aligning geometry across different viewpoints and is particularly sensitive to errors in estimated relative poses (see Left of Figure 2). To overcome this issue, we introduce a pose-aware feature fusion strategy that estimates a unified, holistic representation from a chosen canonical view (see Right of Figure 2). This representation aligns both geometry and appearance across all input views by integrating pose priors into the fusion process.

A key insight of our method is that embedding estimated relative camera rays into multi-view features provides effective guidance to mitigate geometric misalignments in the latent space. Previous works (Zhou & Tulsiani, 2023; Gao et al., 2024; Chen et al., 2024a; Tang et al., 2025) have used ground-truth camera poses with ray-based representations to guide view-conditional generation or 3D reconstruction. However, our approach is distinct in that it operates in a pose-free setting, where relative poses are estimated on the fly. Specifically, we jointly predict Plücker rays and 3D Gaussians in a feed-forward manner directly from input images, eliminating the need for ground-truth pose annotations.

Our ray guidance enables cost aggregation to refine multi-view cost volumes using ray embeddings derived from the estimated poses and provide rich pose information for the reconstruction model. These ray embeddings are injected throughout the refinement process, guiding the framework to build a robust canonical representation from the multi-view cost volumes. Additionally, our *anchor*-based Gaussian prediction estimates fine scene details from arbitrary views based on the fused canonical cost volume. This prediction strategy reduces misalignment by allowing multi-view features to implicitly contribute to Gaussian estimation in a fixed canonical space, rather than estimating Gaussians per view. Further details of each method are provided in the following section.

# 4 MODEL ARCHITECTURE

## 4.1 PROBLEM DEFINITION

SHARE takes $M$ unposed images $I = \{I_i\}_{i=1}^M$ as input. The goal of the model is to learn a mapping function $\Phi_\theta$ that jointly estimates both the relative camera poses (comprising rotation $\mathbf{R} \in SO(3)$ and translation $\mathbf{t} \in \mathbb{R}^3$) and a set of 3D Gaussian primitives $\{G_n\}_{n=1}^N$ using learnable parameters $\theta$. Each Gaussian primitive is defined by its position $\boldsymbol{\mu}_n$, opacity $\alpha_n$, covariance matrix $\boldsymbol{\Sigma}_n$, and color $\mathbf{c}_n$, where the color is represented using spherical harmonics coefficients. We additionally predict a collection of Plücker rays for relative pose representation instead of directly predicting the global camera parameters $\{\mathbf{R}, \mathbf{t}\}$. The Plücker ray representation is formulated locally for patches, where each patch corresponds to a subdivided region of the entire image. Within each patch, the ray is characterized by direction $\mathbf{d} \in \mathbb{R}^3$ and momentum $\mathbf{m} \in \mathbb{R}^3$ vectors, denoted as $\mathbf{P} = (\mathbf{d}, \mathbf{m}) \in \mathbb{R}^6$. Overall, the mapping function can be denoted as follows:

$$\Phi_\theta : \{I_i\}_{i=1}^M \mapsto \left( \{(\boldsymbol{\mu}_n, \alpha_n, \boldsymbol{\Sigma}_n, \mathbf{c}_n)\}_{n=1}^N, \{\mathbf{R}_i, \mathbf{t}_i\} = \Psi(\{\mathbf{P}_i^l\}_{l=1}^{P_h \times P_w}) \right), \quad (1)$$

where $\Psi$ denotes the conversion function from rays to camera parameters, and $P_h \times P_w$ represents the resolution of the patch. The ray representation can be converted to the camera pose by finding the camera center as the closest intersection point of rays and the rotation matrix as the transformation matrix from the predicted ray direction to an identity matrix. We refer to RayDiffusion (Zhang et al., 2024) for details on the conversion between camera and Plücker ray representations.

## 4.2 RAY-GUIDED MULTI-VIEW FUSION

We aim to integrate features from multiple unposed images into a single canonical volume. Each input view offers a unique perspective of the scene in these settings, capturing different parts of geometry and appearance. Without accurate camera pose, aligning these views in 3D space becomes a significant challenge.

To address this, our method jointly estimates relative poses represented as bundles of Plücker rays, which provides geometric guidance during the fusion process. By incorporating pose awareness into both the cost volume construction and the fusion strategy, we ensure that features from different views are coherently aligned in the canonical space. This approach not only enhances geometric coherence but also provides well-aligned features for subsequent Gaussian prediction.

**Joint Feature Extraction.** Recent studies have shown that transformer-based architectures are highly effective in multi-view feature matching for 3D understanding tasks (Li et al., 2021; Ding et al., 2022; Xu et al., 2022; Na et al., 2024; Chen et al., 2024b). We adopt a matching transformer to jointly estimate multi-view features and relative camera poses. In the fusion process, one of the input views is selected as the canonical space, with its local coordinates serving as the reference for the

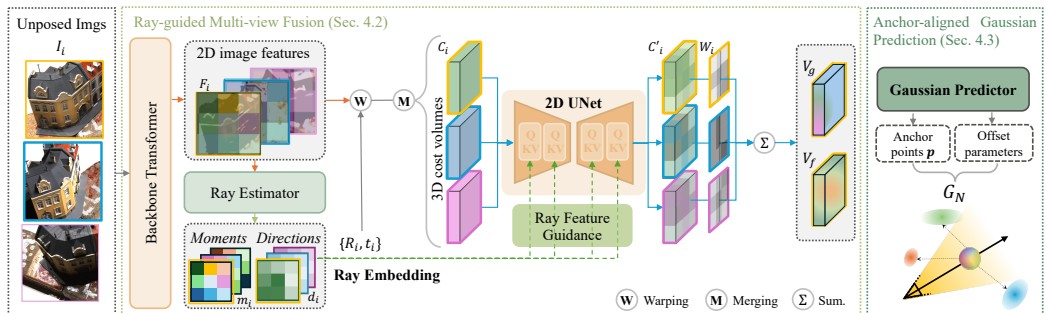

Figure 3: **SHARE Overview.** SHARE consists of two main modules: multi-view fusion (Section 4.2) and anchor-aligned Gaussians prediction (Section 4.3). We first predict camera poses as rays, leveraging features from the shared backbone transformer. Ray embedding provides robust pose guidance during cost volume construction, enhancing the accuracy of multi-view fusion. These pose-aware cost volumes, integrated in the canonical space, guide the Gaussian predictor to estimate positions that ensure consistent geometry across views while preventing potential geometry misalignment. Our anchor-aligned Gaussian prediction module is closely integrated with the pose-aware features, enabling both high-fidelity geometry and detailed shape reconstruction.

others. The output multi-view features are used to estimate patch-wise Plücker rays $\mathbf{P}_i \in \mathbb{R}^{P_h \times P_w \times 6}$ via an additional lightweight two-layer transformer.

**Pose-aware Cost Volumes.** Recent studies in Multi-View Stereo (MVS) have demonstrated that computing correlations among input images enhances robustness across diverse camera configurations (Ding et al., 2022; Chen et al., 2023; Na et al., 2024; Hong et al., 2024). Building on these insights, we extend 2D features into 3D space along hypothetical planes and project them onto other views using predicted poses. This projection process involves converting predicted rays into camera parameters $[\hat{\mathbf{R}}_I, \hat{\mathbf{t}}_I] = \Psi(\hat{r}_i)$, where $\hat{\mathbf{R}}_I$ and $\hat{\mathbf{t}}_I$ denote the predicted camera rotation and translation, respectively. Next, we compute a channel-wise correlation between the reference feature $F_i \in \mathbb{R}^{\frac{H}{4} \times \frac{W}{4} \times C}$ and the warped features for all depth candidates $\{F_d^{j \to i}\}_{d=1}^{D} \in \mathbb{R}^{\frac{H}{4} \times \frac{W}{4} \times D \times C}$, which are obtained by projecting features from view $j$ to view $i$. The resulting cost volume for each view is calculated as:

$$\mathbf{C}_i = \frac{\sum_{j \neq i} F_i \cdot F^{j \to i}}{\sqrt{C}} \in \mathbb{R}^{\frac{H}{4} \times \frac{W}{4} \times D}, \quad \forall i \in \{1, 2, \ldots, M\}. \tag{2}$$

After constructing the cost volumes, we refine them through cost aggregation conditioned on predicted Plücker rays, using patch-wise cross-attention to embed pose awareness into the volumes (Hong et al., 2024; Chen et al., 2024b). We argue that Plücker rays offer several advantages in pose-free, generalizable pipelines. Being locally defined in the 2D spatial dimension, they integrate seamlessly with spatial image features, and their over-parameterized nature introduces a geometric bias that global extrinsic camera matrices cannot capture (Schops et al., 2020). Furthermore, their scale-invariant representation enables effective fusion across varying object scales and camera positions (Chen et al., 2024a).

Specifically, we refine the cost volumes using a transformer-based 2D U-Net with cross-attention layers, where the rays serve as key-value pairs and the cost volumes act as queries. This allows the estimated ray embeddings to provide geometric guidance for constructing a unified canonical volume, with the rays functioning as global positional embeddings across multi-view inputs.

**Canonical Volume Construction.** Capturing the same region often leads to varied observations due to view-dependent effects such as occlusion, lighting, and other environmental factors. We assign spatial weights to each cost volume to account for these variations. The output of the cost aggregation step is a pair of pose-aware cost volumes per view, $\mathbf{C}'_i$, and their corresponding weights, $\mathbf{W}_i$. We then fuse these weighted cost volumes to construct a unified canonical geometry volume, $\mathbf{V}_g$, using a weighted sum. To avoid the trivial solution where one view's weight dominates the fusion process, we add a mean and variance-based volume (Yao et al., 2018) as:

$$\mathbf{V}_g = \sum_{i=1}^{M} (\mathbf{W}_i \cdot \mathbf{C}'_i) + \phi \left( \frac{(\mathbf{C}'_i - \overline{\mathbf{C}}'_i)^2}{M} \oplus \overline{\mathbf{C}}'_i \right) \in \mathbb{R}^{\frac{H}{4} \times \frac{W}{4} \times C}, \tag{3}$$

where $\overline{\mathbf{C}}'$ indicates the average volume, $\oplus$ denotes channel-wise concatenation, and $\phi$ denotes a single CNN layer for channel projection. This multi-view aggregation process enables robust integration of multiple viewpoints, resulting in a unified 3D representation that generalizes effectively across different camera configurations. Similarly, the feature volume $\mathbf{V}_f \in \mathbb{R}^{H \times W \times C}$ is constructed using the same aggregation strategy with the upsampled multi-view features, which learns holistic appearance.

### 4.3 ANCHOR-ALIGNED GAUSSIANS PREDICTION

We extend the pixel-aligned geometry estimation approach, where each pixel is associated with one or more Gaussians (Charatan et al., 2024; Chen et al., 2024b; Wewer et al., 2024). In contrast to previous methods that accumulate geometry predictions from each viewpoint, our approach enables fine geometry prediction within a single canonical view. Our two-stage Gaussians prediction framework first generates sparse geometry anchors, followed by fine-grained dense Gaussian prediction. The overall process follows Figure 4.

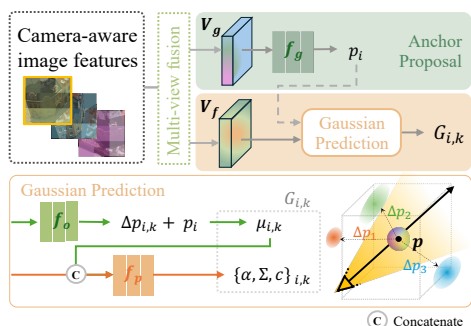

Figure 4: **Gaussians Prediction.** For each pixel, we predict $i$ anchor positions from the geometry volume $V_g$. Subsequently, $k$ Gaussians are estimated to represent the neighboring region.

**Pixel-aligned Anchor Proposal.** Scene geometry exhibits significant variation across different regions. Smooth surfaces such as walls can be adequately represented with larger Gaussians, while intricate textures or complex geometries require smaller, more localized Gaussians. To account for this variability, we introduce a hierarchical approach, beginning with the prediction of anchor points in the canonical space. These anchors serve as geometric centers for Gaussians, enabling adaptive representation for diverse regions. For each pixel in the canonical view, our geometry volume $\mathbf{V}_g$ predicts a 3D anchor position $\mathbf{p} \in \mathbb{R}^{H \times W \times 3}$ by pixel-aligned depth estimation. The depth is predicted by the weighted sum of depth candidates $\mathcal{G} \in \mathbb{R}^{H \times W \times D}$, where the weight is computed with a lightweight MLP depth head $f_g$ and softmax function. This forms the initial structural representation of the scene:

$$\mathbf{p}_i = \mathbf{o} + \mathbf{z}_i \mathbf{d}_i, \quad \mathbf{z}_i = \text{softmax}(f_g(\mathbf{V}_g^i)) \cdot \mathcal{G}, \tag{4}$$

where $\mathbf{p}_i$ is the 3D anchor position for pixel $i$, $\mathbf{d}_i$ is the ray direction and the $\mathbf{z}_i$ is the predicted depth.

**Dense Gaussians Prediction.** The spatial constraints of anchor points within their canonical pixel-aligned, ray-bounded space can lead to suboptimal rendering outcomes due to the sparsity of the coarse geometric representation. To overcome this limitation, we estimate $K$ offset Gaussians, expanding the geometric estimation to a point-wise three-dimensional space, providing greater spatial flexibility and precision. At this stage, we predict the detailed positions, colors, opacities, scalings, and rotations of the offset Gaussians. To capture finer image details, we construct a canonical volume with an upscaled multi-view features as in 3, predicting feature volume $\mathbf{V}_f$ to incorporate finer information. The geometry channels of $\mathbf{V}_f$ are passed through the offset prediction MLP head $f_o$, which predicts the offset vectors $\Delta \mathbf{p}_k = f_o(\mathbf{V}_f)$, for the Gaussian positions. These offset vectors are then concatenated with the remaining channels of $\mathbf{V}_f$ Another MLP head, $f_p$, processes the concatenated features to estimate the remaining Gaussian parameters. Consequently, the Gaussian prediction for each anchor is represented as:

$$\mathbf{G}_{i,k} = \{\boldsymbol{\mu}, \alpha, \boldsymbol{\Sigma}, \mathbf{c}\}_{i,k}, \quad \mu_{i,k} = \mathbf{p}_i + \Delta \mathbf{p}_{i,k}, \quad \{\alpha, \boldsymbol{\Sigma}, \mathbf{c}\}_{i,k} = f_p(\text{PE}(\Delta \mathbf{p}_{i,k}), \mathbf{V}_f^i) \tag{5}$$

where $\Delta \mathbf{p}_{i,k}$ represents the offset for the $k$-th Gaussian relative to anchor $\mathbf{p}_i$, $\text{PE}(\cdot)$ denotes positional embedding, and $\mathbf{G}_{i,k}$ encompasses the set of Gaussian parameters. This formulation provides flexibility in capturing spatial details by adaptively positioning Gaussians near the anchor point.

### 4.4 TRAINING AND INFERENCE

SHARE is trained using a combination of photometric and ray regression losses. The photometric loss includes Mean Squared Error (MSE) and Learned Perceptual Image Patch Similarity (LPIPS) (Zhang et al., 2018):

$$\mathcal{L}_{\text{total}} = \lambda_{\text{MSE}} \cdot \sum_{i=1}^{M} \left\| \hat{\mathcal{I}}_i - \mathcal{I}_i \right\|_2^2 + \lambda_{\text{LPIPS}} \cdot \text{LPIPS}(\hat{I}, I_{\text{gt}}) + \lambda_{\text{ray}} \cdot \sum_{i=1}^{N} \left\| \hat{\mathcal{R}}_i - \mathcal{R}_i \right\|_2^2,$$

where $\hat{\mathcal{I}}_i$ and $\mathcal{I}_i$ represent the predicted and ground truth images, respectively, and LPIPS measures perceptual similarity between the predicted image $\hat{I}$ and ground truth $I_{\text{gt}}$. $\hat{\mathcal{R}}_i$ and $\mathcal{R}_i$ denote the predicted and ground truth rays for ray regression. The loss terms are weighted by $\lambda_{\text{MSE}}$, $\lambda_{\text{LPIPS}}$, and $\lambda_{\text{ray}}$. Once trained, our model only requires RGB images as input to reconstruct the scene and estimate the relative poses.

## 5 EXPERIMENTAL RESULTS

### 5.1 EXPERIMENTAL SETTINGS

**Datasets.** We train and evaluate our method on two distinct real-world datasets, DTU (Jensen et al., 2014) and RealEstate10K (Zhou et al., 2018). These datasets were selected to assess our approach across varying scene scales and camera configurations.

*DTU (Jensen et al., 2014)* contains small-scale static scenes captured from 49 cameras with diverse configurations, provided with camera parameters. We use 75 scenes for training and 15 for testing (Na et al., 2024). This dataset validates SHARE on small-scale scenes with diverging camera baselines.

*RealEstate10K (Zhou et al., 2018)* contains 67,477 training and 7,289 testing scenes from YouTube real estate videos, with camera parameters for each frame. We follow the train and test splits of previous work (Chen et al., 2024b). This dataset, featuring typical camera movements in real-world videos, evaluates SHARE on large-scale data.

Furthermore, we conducted cross-dataset experiments by evaluating SHARE on the ACID (Liu et al., 2021) and BlendedMVS (Yao et al., 2020) datasets, each trained using the RealEstate10K (Zhou et al., 2018) and DTU (Jensen et al., 2014) datasets. Detailed descriptions of the datasets and the evaluation results are provided in Appendix A.4.

**Baselines.** We compare our method with state-of-the-art generalizable 3D Gaussian splatting (g-3DGS) approaches, including PixelSplat (Charatan et al., 2024) and MVSplat (Chen et al., 2024b), as well as pose-free generalizable NeRF methods, including LEAP (Jiang et al., 2023), CoPoN-eRF (Hong et al., 2024), and FlowCAM (Smith et al., 2023). For g-3DGS, we evaluate performance under different pose settings, including poses predicted by one of the state-of-the-art pose prediction, DUSt3R (Wang et al., 2024), and poses corrupted with random Gaussian noise, following the protocol in (Truong et al., 2023). For generalizable NeRFs, we compare our approach to LEAP (Jiang et al., 2023) on the DTU dataset and the others (Hong et al., 2024; Smith et al., 2023) on the RealEstate10K dataset, where each method is evaluated from their studies. Further details of our baselines can be found in the Appendix A.2.

**Metrics.** To assess the quality of our method, we employ commonly used metrics in the field of novel view synthesis, including Peak Signal-to-Noise Ratio (PSNR), Structural Similarity Index (SSIM) (Wang et al., 2004), and Learned Perceptual Image Patch Similarity (LPIPS) (Zhang et al., 2018). Additionally, we measure the accuracy of relative camera positioning with Rotation and Translation errors. Specifically, we use geodesic rotation error and angular difference for translation following (Hong et al., 2024).

**Implementation Details.** We trained our model on the DTU dataset using three context images and one target image, and on the RealEstate10K dataset with two context images and three target images, following standard protocols. Both datasets used an image resolution of $224 \times 224$. Our multi-view fusion backbone consists of six matching Transformer layers (Chen et al., 2024b), while dense ray prediction is handled by a modified 2-layer Transformer model (Zhang et al., 2024). The resolution

Table 1: **Quantitative results on DTU datasets.** We compare our method with generalizable Gaussian splatting and pose-free generalizable NeRFs. We select small and large following view selection score (Yao et al., 2018). Pred* denotes the estimated pose using DUSt3R (Wang et al., 2024). We highlight the **best** and second-best results. The results from generalizable Gaussian splatting with ground truth given are colored as gray.

| Method | Pose | | | Small baseline | | | | | Larger baseline | | |
|---|---|---|---|---|---|---|---|---|---|---|---|
| | | Rot. ↓ | Trans. ↓ | PSNR ↑ | SSIM ↑ | LPIPS ↓ | Rot. ↓ | Trans. ↓ | PSNR ↑ | SSIM ↑ | LPIPS ↓ |
| PixelSplat | GT | – | – | 20.96 | 0.65 | 0.31 | – | – | 19.64 | 0.62 | 0.33 |
| | $\sigma = 0.01$ | 1.01 | 1.71 | 16.84 | 0.43 | 0.47 | 1.01 | 0.83 | 16.60 | 0.44 | 0.46 |
| | $\sigma = 0.03$ | 3.03 | 5.15 | 14.31 | 0.34 | 0.61 | 3.04 | 3.03 | 14.43 | 0.36 | 0.57 |
| | $\sigma = 0.05$ | 5.05 | 8.61 | 13.19 | 0.31 | 0.65 | 5.06 | 3.99 | 13.59 | 0.33 | 0.61 |
| | Pred* | 1.77 | 13.66 | 15.98 | 0.42 | 0.47 | 2.87 | 13.96 | 15.35 | 0.41 | 0.51 |
| MVSplat | GT | – | – | 21.00 | 0.69 | 0.24 | – | – | 19.82 | 0.63 | 0.28 |
| | $\sigma = 0.01$ | 1.01 | 1.71 | 16.43 | 0.42 | 0.42 | 1.01 | 0.83 | 16.43 | 0.42 | 0.42 |
| | $\sigma = 0.03$ | 3.03 | 5.15 | 13.74 | 0.32 | 0.55 | 3.04 | 3.03 | 13.90 | 0.33 | 0.54 |
| | $\sigma = 0.05$ | 5.05 | 8.61 | 12.78 | 0.28 | 0.60 | 5.06 | 3.99 | 12.90 | 0.29 | 0.59 |
| | Pred* | 1.77 | 13.66 | 13.22 | 0.32 | 0.58 | 2.87 | 13.96 | 14.69 | 0.35 | 0.52 |
| LEAP | – | – | – | 18.76 | 0.54 | 0.48 | – | – | 17.77 | 0.51 | 0.48 |
| Ours | – | 2.74 | 6.28 | **19.94** | **0.63** | **0.28** | 6.85 | 5.84 | **18.78** | **0.58** | **0.34** |

of the anchor-aligned Gaussian prediction matches the image resolution, with each anchor associated with three Gaussian primitives ($N = K * H * W$). The training was performed for 140,000 iterations on DTU and 300,000 on RealEstate10K using the Adam optimizer. All experiments were run on an NVIDIA RTX 4090 GPU. Additional implementation details are provided in the Appendix A.3.

## 5.2 NOVEL VIEW SYNTHESIS.

To evaluate its performance, we conduct experiments on the DTU (Jensen et al., 2014) and RealEstate10K (Zhou et al., 2018) datasets, which encompass a range of scenarios from object-centric indoor scenes to large-scale environments.

On the DTU dataset (Table 1), where scenes are captured with large camera transformations around a central object, our method exhibits superior performance across varying camera baselines. Notably, we outperform g-3DGS methods even with slightly noisy pose conditions. In particular, our approach achieves LPIPS (Zhang et al., 2018) and SSIM (Wang et al., 2004) scores comparable to those of g-3DGS with ground-truth poses, highlighting the robustness of our pose-aware multi-view fusion approach in handling pose-free scenarios with diverse camera configurations.

Our key observation is that g-3DGS baselines, when using noisy poses, often produce significantly distorted renderings. This emphasizes the sensitivity of g-3DGS methods to pose inaccuracies, especially in sparse-view settings where even minor pose noise (e.g., $\sigma = 0.01$, with less than $1°$ angular error) leads to noticeable degradation in performance.

On the RealEstate10K dataset (Table 2), which features larger-scale scenes, our method consistently outperforms both g-3DGS with estimated poses and pose-free generalizable NeRF methods (Hong et al., 2024; Smith et al., 2023). These results further demonstrate the scalability and adaptability of our approach across different scene complexities.

Figure 5 provides qualitative visualizations of rendered results, showing superior fidelity and geometric consistency of our method compared to the baselines on both datasets. These visual results further validate the effectiveness of our approach in challenging, pose-free scenarios.

**Geometry Reconstruction.** The strength of our method is particularly evident when visualizing the reconstructed geometry through rendered depth maps (see Figure 6). On both the DTU and RealEstate10K datasets, SHARE consistently predicts accurate geometry, while baselines often produce incorrect and noisy geometry, even when their synthesized colors appear acceptable. Notably, CoPoNeRF (Hong et al., 2024), despite its high rendering performance, frequently fails in depth prediction. This reflects SHARE's ability to accurately capture appearance and geometry, thanks to the ray-guided multi-view fusion pipeline.

## 5.3 ABLATION STUDY

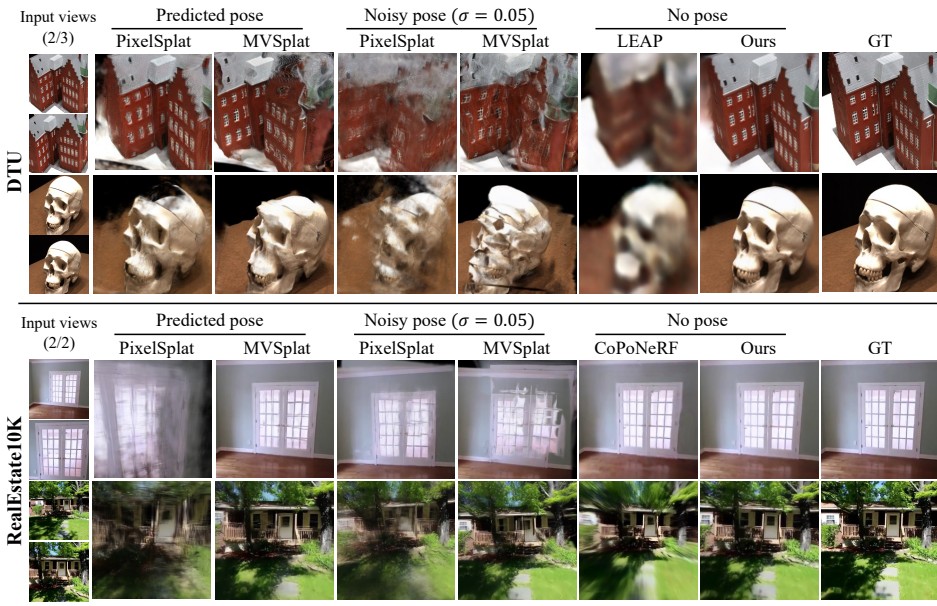

Figure 5: **Qualitative results on DTU and RealEstate10K datasets.** We visualized rendering results of multiple scenes from DTU and RealEstate10K datasets. Our method captures fine details with correct geometry. More qualitative results can be found in the Appendix.

Table 2: **Quantitative results on RealEstate10K Datasets.** We compare the results of novel view synthesis on large-scale datasets. Pred* denotes the estimated pose using DUSt3R (Wang et al., 2024).

| Method | Pose | Rot. ↓ | Trans. ↓ | PSNR ↑ | SSIM ↑ | LPIPS ↓ |
|---|---|---|---|---|---|---|
| PixelSplat | GT | – | – | 26.08 | 0.86 | 0.14 |
| | $\sigma = 0.01$ | 0.92 | 1.56 | 20.14 | 0.62 | **0.23** |
| | $\sigma = 0.03$ | 2.75 | 4.70 | 17.07 | 0.50 | 0.38 |
| | $\sigma = 0.05$ | 4.58 | 7.65 | 15.69 | 0.46 | 0.46 |
| | Pred* | 1.76 | 12.20 | 11.73 | 0.34 | 0.60 |
| MVSplat | GT | – | – | 26.39 | 0.87 | 0.13 |
| | $\sigma = 0.01$ | 0.92 | 1.56 | 19.99 | 0.62 | **0.23** |
| | $\sigma = 0.03$ | 2.75 | 4.70 | 16.67 | 0.48 | 0.37 |
| | $\sigma = 0.05$ | 4.58 | 7.64 | 15.12 | 0.44 | 0.45 |
| | Pred* | 1.76 | 12.20 | 17.81 | 0.56 | 0.33 |
| FlowCAM | – | 7.43 | 50.66 | 18.24 | 0.60 | 0.64 |
| CoPoNeRF | – | 3.61 | 12.77 | 19.54 | 0.40 | 0.64 |
| Ours | – | 3.78 | 11.61 | **21.23** | **0.71** | 0.26 |

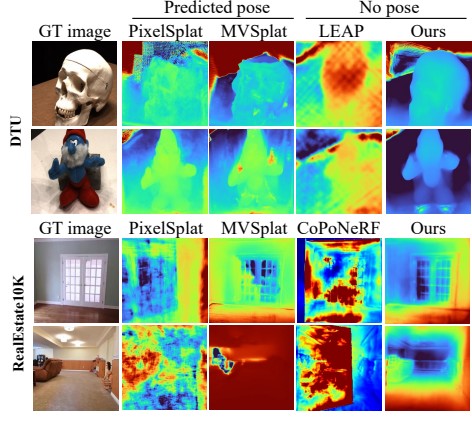

Figure 6: **Rendered depth comparison.** We compare our rendered depth with other baselines. Ours shows robust geometry reconstruction, while others fail to capture the correct depth.

**Pose Embedding.** We hypothesize that incorporating per-pixel pose priors into the reconstruction pipeline enhances geometric consistency by improving spatial awareness during 3D geometry estimation. Quantitative results in Table 3 demonstrate a significant decrease in quality metrics when pose embedding is absent. Fig. 7 illustrates that training without pose embedding leads to severe geometric distortions, particularly evident in viewpoints divergent from the canonical view. The marked improvement in the depth map with pose embedding suggests that pose-aware features effectively encode geometric details across multiple views and their inter-relationships.

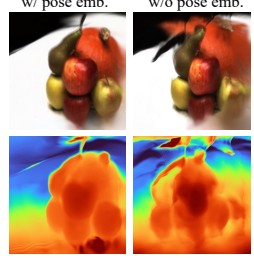

Figure 7: **Effect of pose embedding.**

**Mean and Variance Volume.** The weighted cost volume in Eq. (3) captures complex visibility information, but the mean-variance volume is crucial for avoiding trivial solutions where a single view dominates the fusion process (Yao et al., 2018). By enhancing train-

Table 3: **Ablation Study on DTU datasets.** We evaluate the impact of pose embedding and the number of offsets on rendering quality using the DTU dataset, considering both small and large baselines and averaging the results. The full model delivers the best performance. Excluding the pose embedding causes a significant drop in rendering quality while increasing the number of offsets consistently improves the results.

| Method | PSNR ↑ | SSIM ↑ | LPIPS ↓ |
|---|---|---|---|
| *w/o pose embedding* | 17.52 | 0.54 | 0.37 |
| *anchor only* | 14.05 | 0.38 | 0.57 |
| *w/ 1 offsets* | 19.11 | 0.60 | 0.32 |
| *w/ 2 offset* | 19.33 | **0.61** | **0.30** |
| *Ours* | **19.36** | **0.61** | 0.31 |

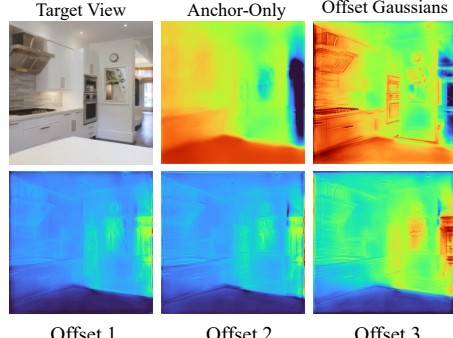

Figure 8: **Analysis of *anchor*-aligned Gaussian prediction on RealEstate10K datasets.** (First row) Anchor estimates the overall coarse structure of the scene, while offsets estimate the finer details. (Second row) Different offsets focus on different layers of depth.

ing stability and balancing contributions across views, it prevents over-reliance on any single view. Table 3 quantifies the performance drop when excluded.

**Anchor-Aligned Gaussians Prediction.** We analyze the efficacy of predicting anchor-based Gaussians by verifying how our offset prediction covers various geometric details near the anchor points. We first compare quantitative rendering quality with varying numbers of offsets, as shown in Table 3. The results demonstrate that dense offset prediction covers fine details as we expected. We further validated this by visualizing $K = 3$ offsets, revealing that each offset is responsible for distinct depth regions (Figure 8). This indicates that our learned holistic representation provides sufficient information for the Gaussian prediction stage to deal with various viewpoints effectively.

# 6 CONCLUSION

In this work, we present SHARE, the first pose-free generalizable 3D Gaussian splatting approach validated across small and large-scale scene-level datasets with varying camera baselines. Our ray-guided multi-view fusion strategy effectively addresses geometry ambiguity caused by incorrect poses, capturing a unified feature representation from all input views into a single canonical estimation space. Our two-stage Gaussian predictor aligns seamlessly with this strategy, successfully capturing fine shape details visible in multi-view data, even within the constrained estimation space. Through comprehensive experiments, we demonstrated that our model can reconstruct high-fidelity 3D structures across small and large scene scales, even in challenging scenarios involving large camera baselines. We believe that our approach introduces a novel approach for effectively addressing geometry misalignment with a pose-aware fusion pipeline.

**Limitations and Discussions.** While SHARE achieves strong pose-free generalization across diverse settings, it is not without its limitations. One potential drawback is its performance in scenes with significantly limited overlap between views. Thus, applying SHARE to challenging scenarios, such as 360-degree scenes with sparse image inputs, presents an area that could benefit from further exploration. Another consideration is the reliance on camera intrinsic parameters, which could restrict the method's usability in practical scenarios involving varied image sources, such as datasets captured with different cameras or online images lacking metadata. Addressing these challenges and broadening the scope of our approach present compelling directions for future work.

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

# A APPENDIX

In this section, we describe the followings:

- Detailed Discussion on Ray Guidance.
- Details of baseline implementation
- Details of model implementation.
- Additional Results.

## A.1 DETAILED DISCUSSION ON RAY GUIDANCE

Most conventional methods in Multi-view Stereo (MVS) utilize cameras to establish geometrical relationships across the input views. However, the relationship becomes unreliable when given poses are noisy. We argue that it is important for the image features to have an awareness of camera poses to mitigate the influence of unreliable relationships during the 3D reconstruction. To this end, we combine predicted Plücker rays with image features to construct the cost volume, leveraging the advantages of using a generic camera representation. The intuition behind our design choice is to inject awareness of camera pose in multi-view space to each image feature.

Specifically, we project features from different viewpoints to compute correlation among input images by converting rays into camera poses and performing homography warping. While this allows some pose error, which leads to misalignment in feature space, we rectify the cost volumes by pose-aware cost aggregation process described in Section 4.2 of the main paper. As shown in Figure 9, eliminating pose embedding leads to large discrepancies in geometry estimation, leading to blurry images or introducing artifacts. This highlights the importance of pose embedding in our fusion process.

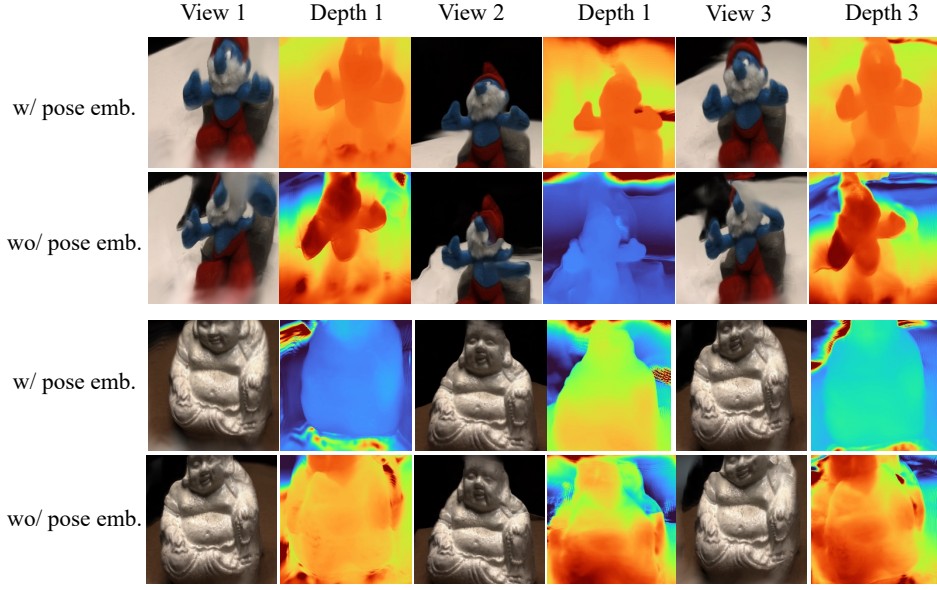

Figure 9: **Additional Qualitative Ablation Results on Pose Embedding.** Estimating geometry without pose embedding results in significant failures, producing blurry artifacts and misaligned structures in the 3D reconstruction. With pose embeddings, SHARE demonstrates the importance of geometric bias, achieving more accurate and sharper reconstructions. This highlights the effectiveness of pose-aware fusion in handling pose errors during the multi-view reconstruction process.

## A.2 DETAILS OF BASELINE IMPLEMENTATION

For the small-scale DTU dataset (Jensen et al., 2014), we compared and validated our method against the pose-free baseline LEAP (Jiang et al., 2023). The LEAP model was trained on the DTU 3-

Table 4: **Comparison on baselines with different pose prediction methods on DTU dataset.**

| Method | Pose | Rot. ↓ | Trans. ↓ | PSNR ↑ | SSIM ↑ | LPIPS ↓ |
|--------|------|--------|----------|--------|--------|---------|
| | GT | – | – | 20.96 | 0.65 | 0.31 |
| | COLMAP | 7.10 | 31.62 | 13.49 | 0.34 | 0.66 |
| PixelSplat | MASt3R | 2.40 | **3.52** | 15.69 | 0.40 | 0.50 |
| | DUSt3R | **1.77** | 13.66 | 15.98 | 0.42 | 0.47 |
| | Ours | 2.74 | 6.28 | 13.29 | 0.31 | 0.66 |
| | GT | – | – | 21.00 | 0.69 | 0.24 |
| | COLMAP | 7.10 | 31.62 | 14.69 | 0.44 | 0.46 |
| MVSplat | MASt3R | 2.40 | **3.52** | 13.31 | 0.31 | 0.58 |
| | DUSt3R | **1.77** | 13.66 | 13.22 | 0.32 | 0.58 |
| | Ours | 2.74 | 6.28 | 14.08 | 0.33 | 0.51 |
| Ours | – | 2.74 | 6.28 | **19.94** | **0.63** | **0.28** |

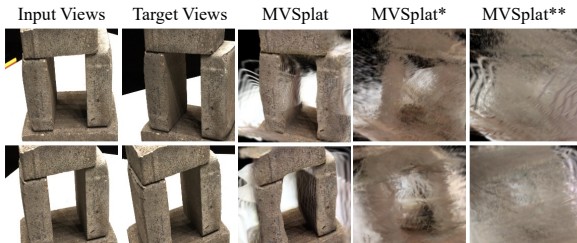

Input Views   Target Views   MVSplat   MVSplat*   MVSplat**

Figure 10: **Rendering of MVSplat Trained with Predicted and Noisy Poses.** The row labeled MVSplat shows the results of training with ground-truth poses, while MVSplat* and MVSplat** refer to the MVSplat model trained with predicted poses from DUSt3R and noisy poses with minor errors, respectively.

Table 5: **Quantitative results of pose estimation performance.** We evaluate the pose estimation performance on DTU dataset with small baselines, given three input views. The lowest error is marked as bold.

| Method | Rot. ↓ | Trans. ↓ |
|--------|--------|----------|
| DUSt3R | 1.77 | 13.66 |
| MASt3R | 2.40 | 3.52 |
| COLMAP | 7.10 | 31.62 |
| Relpose++ | 19.56 | 44.18 |
| RayRegression | 3.10 | 6.57 |
| Ours | **2.74** | **6.28** |

view dataset for 140K iterations. Since our evaluation on DTU uses three input views, we also trained pose-dependent state-of-the-art generalizable 3D reconstruction methods, including Pixel-Splat (Charatan et al., 2024) and MVSplat (Chen et al., 2024b), with a batch size of 1 for 140K iterations.

For the large-scale RealEstate10K dataset (Zhou et al., 2018), we compared our method against pose-free baselines CoPoNeRF (Hong et al., 2024) and FlowCam (Smith et al., 2023). Since Co-PoNeRF and FlowCam use the same train-test split as our method, we directly compared our results with the reported values. Additionally, PixelSplat and MVSplat were evaluated using their pretrained checkpoints on the same 2-view train-test split settings.

We evaluated pose-dependent baselines under two conditions: using predicted poses and poses perturbed by random noise. For predicted poses, we used one of the state-of-the-art pose estimator, DUSt3R (Wang et al., 2024), to estimate poses from the input images. To ensure fair comparisons, we also evaluated the baselines with various pose estimators, including COLMAP (Schonberger & Frahm, 2016), DUSt3R (Wang et al., 2024), MASt3R (Leroy et al., 2024) and SHARE. For DUSt3R and MASt3R, we utilized pre-trained model weights provided in their official GitHub repositories. As shown in Table 4, our method consistently outperformed these combinations. Furthermore, the results for noisy poses (Table 1 and Table 2) highlight that even minor errors—currently unavoidable by state-of-the-art pose estimators—can introduce significant instability in reconstruction quality.

We trained the baseline models using ground-truth (GT) poses, as training with noisy poses lacking specific noise patterns often resulted in instability, divergence, or failure to converge. Figure 10 illustrates a comparison of MVSplat models trained on DTU with GT poses versus those trained with predicted poses from DUSt3R (Wang et al., 2024) and slightly perturbed poses ($\sigma = 0.01$, rotation error 0.95°, translation error 1.05°). These findings demonstrate that even small amounts of noise during training can destabilize models by introducing subtle misalignments between views, leading to a decline in reconstruction quality.

A.3 DETAILS OF MODEL IMPLEMENTATION

In this section, we'll discuss our framework in more detail. Given sparse-view unposed images, our goal is to build comprehensive Gaussians in a canonical space. The output of the multi-view feature extractor is $V \times C \times H \times W$, where we set $C$ as 128 in all experiments. Given these features, we estimate the relative Plücker rays $V \times 6 \times H \times W$ with two additional transformer blocks following the U-Net structure of (Zhang et al., 2024). Then, we embed ray with a lightweight MLP to latent space and modulate multi-view features using AdaLN (Peebles & Xie, 2023), following LaRa (Chen et al., 2024a). In the ray-guided multi-view fusion process, we first build the cost volumes from all input views, where the depth candidates $D$ are all set to 128. We warp all the features to the reference views with the estimated pose (converted from Plücker rays). Then, we build the geometry volume $V_g$ as in 3. The geometry volume is used to estimate the anchor points $3 \times \frac{H}{4} \times \frac{W}{4}$. Simultaneously, we build the feature volume $V_f$ in a similar manner, but with the upscaled multi-view features, to estimate the offset vectors and Gaussian parameters necessary for finer detail reconstruction.

We divide channels of $\mathbf{V}_f$ for displacement prediction of anchor points (32), and the remaining channels (96) encode texture-related Gaussian parameters. The geometry channels of $\mathbf{V}_f$ are passed through the offset prediction MLP head $f_o$, which predicts the offset vectors $\Delta \mathbf{p}_k = f_o(\mathbf{V}_f)$, for the Gaussian positions. We set $K = 3$ for all experiments. These offset vectors are then concatenated with the remaining channels of $\mathbf{V}_f$ Another MLP head, $f_p$, processes the concatenated features to estimate the remaining Gaussian parameters.

A.4 ADDITIONAL RESULTS

**Results of pose estimation** We evaluated our pose estimation performance in terms of rotation error (degrees) and translation error (degrees), as detailed in the main paper. Comparisons were made against state-of-the-art pose estimators, including DUSt3R (Wang et al., 2024), MASt3R (Leroy et al., 2024), and RayRegression from Cameras-as-Rays (Zhang et al., 2024). Additionally, we compared our method with COLMAP (Schonberger & Frahm, 2016) for primitive pose estimation and RelPose++ (Lin et al., 2024) as a direct 6D pose estimator. The evaluation used three small-baseline views from the DTU (Jensen et al., 2014) dataset as input images.

While our primary objective is high-quality novel view synthesis rather than pose estimation, our method achieves pose estimation performance comparable to state-of-the-art methods, further demonstrating its robustness and versatility.

**Cross-dataset generalization** Table 6 and Figure 11 present the cross-dataset generalization results, comparing our proposed method, SHARE, with baseline approaches. Models trained on the RealEstate10K (Zhou et al., 2018) dataset were evaluated on the ACID (Liu et al., 2021) dataset, while those trained on the DTU (Jensen et al., 2014) dataset were tested on BlendedMVS (Yao et al., 2020). The ACID dataset comprises natural large-scaled scenes captured using aerial drones, divided into 11,075 scenes for training and 1,972 scenes for testing, with accompanying camera extrinsic and intrinsic parameters. The BlendedMVS dataset consists of 3D models of diverse scenes, including outdoor and indoor environments. In our experiments, we utilize a subset of BlendedMVS as a cross-dataset evaluation benchmark to assess the generalization ability of our method.

Notably, under the challenging conditions of pose error $\sigma = 0.01$, which remains difficult even for state-of-the-art pose estimators, SHARE consistently outperforms all baseline methods across all metrics. These findings underscore the robustness of SHARE, particularly in realistic scenarios where pose estimation inaccuracies are inevitable.

**Comparision with the Concurrent Work.** We compare SHARE with our concurrent work, Splatt3R (Smart et al., 2024) which utilizes pretrained MASt3R (Leroy et al., 2024) weights for geometry estimation. Since Splatt3R requires ground-truth dense depths map during training, it is not directly applicable to our used datasets (RealEstate10K (Zhou et al., 2018) doesn't contain gt depths, and DTU (Jensen et al., 2014) contains masked depths, which we found that it is not directly applicable without method modifications because of Splatt3R's pixel-aligned dense prediction mechanism). Instead, we directly compare with the pretrained Splatt3R model trained on ScanNet++ (Yeshwanth et al., 2023). We note that Splatt3R employs a "masking loss" (refer to Section 3.4 in their paper) to render only valid pixels for the target view based on input images. To avoid this issue, we measure PSNR and other metrics only for the valid pixels produced by Splatt3R (pixels with $> 0$ values).

Table 6: **Quantitative comparison of cross-dataset generalization.** The best-performing values across all metrics are highlighted in bold.

| Method | Pose | RealEstate10K → ACID | | | DTU → BlendedMVS | | |
| | | PSNR↑ | SSIM↑ | LPIPS↓ | PSNR↑ | SSIM↑ | LPIPS↓ |
|---|---|---|---|---|---|---|---|
| pixelSplat | GT | 26.84 | 0.81 | 0.18 | 11.64 | 0.20 | 0.67 |
| | $\sigma = 0.01$ | 21.73 | 0.57 | 0.28 | 11.65 | 0.20 | 0.68 |
| MVSplat | GT | 28.18 | 0.84 | 0.15 | 12.04 | 0.19 | 0.56 |
| | $\sigma = 0.01$ | 21.65 | 0.57 | 0.27 | 11.92 | 0.20 | 0.59 |
| Ours | - | **23.47** | **0.69** | **0.26** | **12.19** | **0.26** | **0.61** |

Table 7: **Quantitative Comparison with Concurrent Work.** We compare our method with the concurrent work Splatt3R on the DTU and RealEstate10K datasets, using two input views for both datasets. Splatt3R results are obtained using pretrained weights trained on the ScanNet++ dataset, while our method is trained on each respective dataset. The best results are highlighted in bold.

| Method | DTU (2-views) | | | RealEstate10K | | |
| | PSNR ↑ | SSIM ↑ | LPIPS ↓ | PSNR ↑ | SSIM ↑ | LPIPS ↓ |
|---|---|---|---|---|---|---|
| Splatt3R | 11.78 | 0.28 | 0.57 | 15.80 | 0.53 | 0.30 |
| Ours | **17.50** | **0.34** | **0.48** | **21.23** | **0.71** | **0.26** |

Including entire regions would lead to significant drops in PSNR and thus would not reflect the method's intended performance.

In Table 7 and Figure 12, we present comparisons both on the DTU and RealEstate10K datasets, where SHARE outperforms Splatt3R. To ensure fairness, as comparing Splatt3R trained on Scan-Net++ with SHARE trained on each dataset may introduce biases, we conducted additional evaluations in a cross-dataset setting. Specifically, we compared Splatt3R trained on ScanNet++ and SHARE trained on RealEstate10K in the ACID (Liu et al., 2021) dataset. As illustrated in Table 8 and Figure 13, SHARE demonstrates superior rendering quality compared to Splatt3R. We measure metrics only for the valid pixels produced by Splatt3R (pixels with $> 0$ values). Including entire regions would lead to significant drops in PSNR and thus would not reflect the method's intended performance. Splatt3R exhibits scale ambiguity in its predicted scenes, which can lead to a substantial drop in performance when applied to datasets with unseen scale distributions.

**Discussion on large baseline inputs** We visualized large-baseline camera scenarios (Figure 14). We compare our method with PixelSplat (Charatan et al., 2024) and MVSplat (Chen et al., 2024b) using both our predicted poses and perturbed poses with Gaussian noise, which exhibit similar or lower pose errors compared to predicted poses.

**Discussion on Efficiency.** We evaluated and compared the inference time (in seconds) and GPU memory usage (in MB) of our method against baseline approaches on the RealEstate10K dataset, as detailed in Table 9. Inference time is measured as the end-to-end duration required for novel view synthesis using two unposed input images, while GPU memory usage includes both static and dynamic memory allocations during inference. Our method achieves superior efficiency in both inference time and GPU memory usage compared to the pose-free, generalizable NVS baseline CoPoNeRF (Hong et al., 2024) and the concurrent method Splatt3R. Furthermore, our approach

Table 8: **Quantitative Comparison with Concurrent Work: Cross-Dataset Generalization.** We evaluate and compare the cross-dataset generalization performance of our method and Splatt3R. The best results are highlighted in bold.

| Method | Training data | ACID | | |
| | | PSNR ↑ | SSIM ↑ | LPIPS ↓ |
|---|---|---|---|---|
| Splatt3R | ScanNet++ | 17.49 | 0.63 | **0.26** |
| Ours | RealEstate10K | **23.47** | **0.69** | **0.26** |

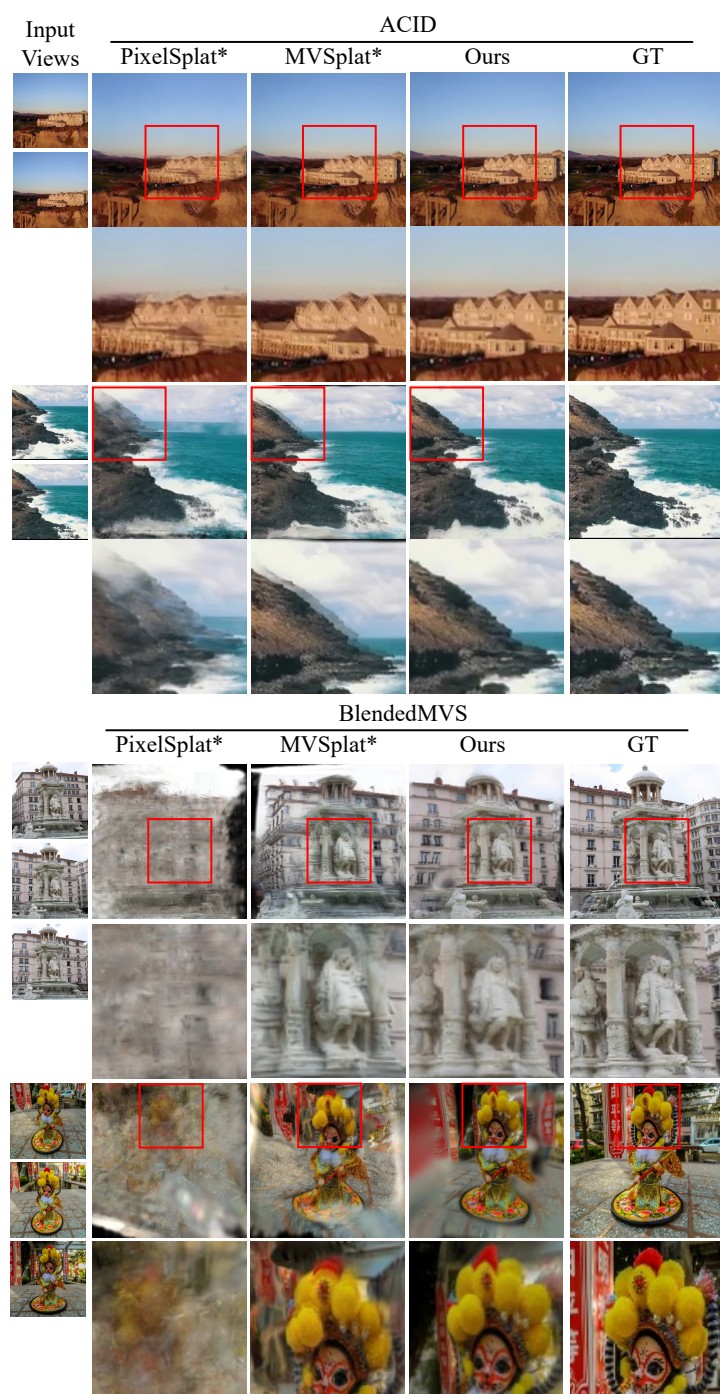

Figure 11: **Qualitative Results for Novel View Synthesis in Cross-Dataset Generalization.** PixelSplat* and MVSplat* denote methods combined with noisy camera settings ($\sigma = 0.01$). To aid visibility, we highlight the regions of interest with red boxes and provide close-up visualizations of these areas for detailed comparison.

delivers the highest rendering quality among the compared methods, underscoring its effectiveness. All experiments were conducted on an NVIDIA RTX 4080 GPU.

**Qualitative results of novel view synthesis** We present our additional results on the DTU (Jensen et al., 2014) dataset (Figure 15) and RealEstate10K (Zhou et al., 2018) dataset (Figure 16).

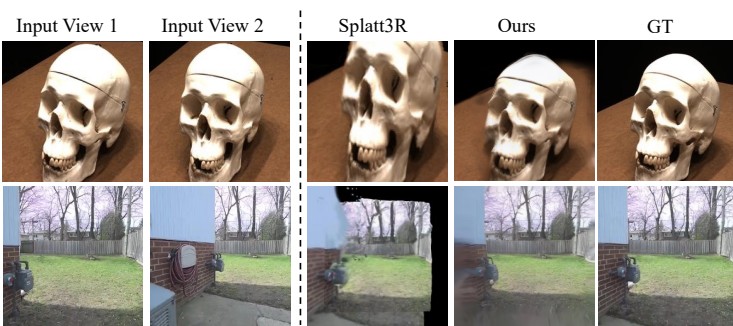

Figure 12: **Qualitative Comparision with the Concurrent Work.**

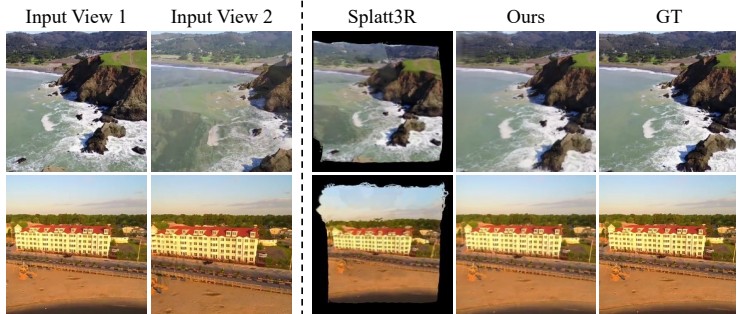

Figure 13: **Qualitative Comparision with the Concurrent Work: Cross-dataset Generalization.**

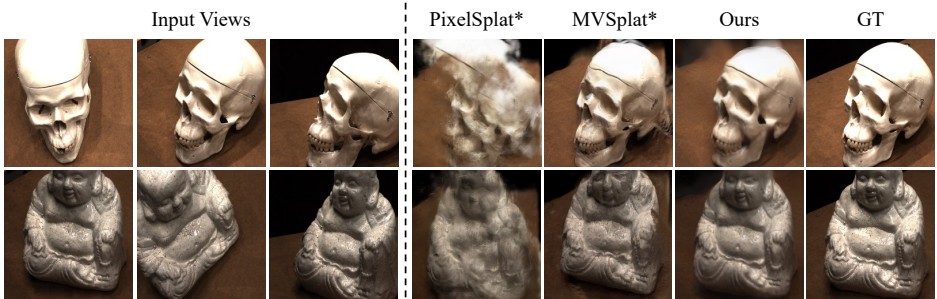

Figure 14: **Qualitative Results of Novel View Synthesis with Large-Baseline View Sets.** PixelSplat and MVSplat denote methods combined with a noisy camera setup, incorporating Gaussian noise with a standard deviation of 0.01.

Table 9: **Model Efficiency Measurements.** Each metric is evaluated across models using the same dataset configuration and averaged for consistency.

| Method | Inference time (s) | GPU Memory (MB) |
|---|---|---|
| CoPoNeRF | 3.37 | 9587.22 |
| MVSplat + MASt3R | 0.22 | **4376.94** |
| Splatt3R | 0.26 | 6198.00 |
| Ours | **0.17** | 5887.18 |

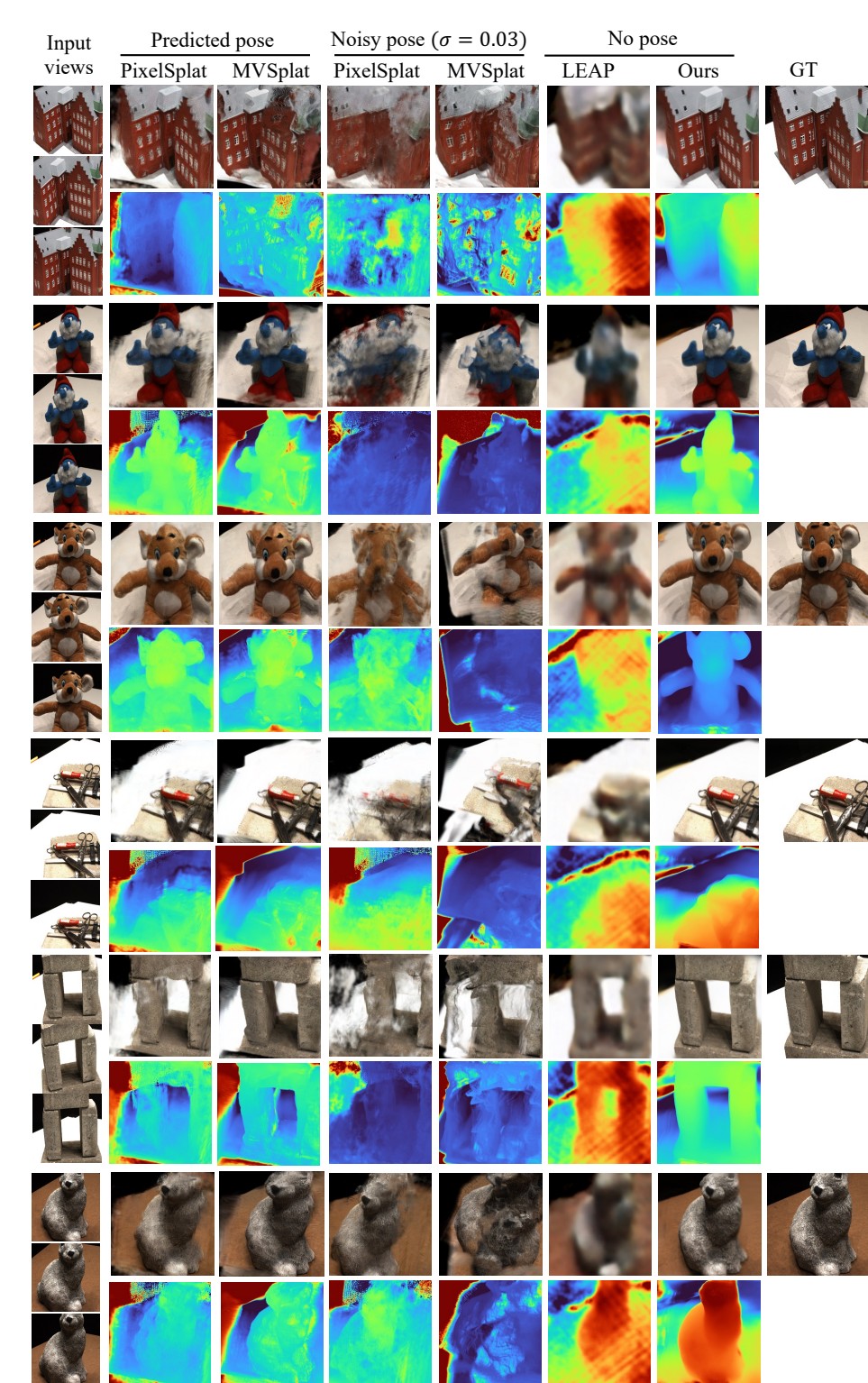

Figure 15: **Additional Qualitative Results on the DTU Dataset.** Rendered target images are shown based on three input views. The predicted pose indicates poses predicted using DUSt3R (Wang et al., 2024).

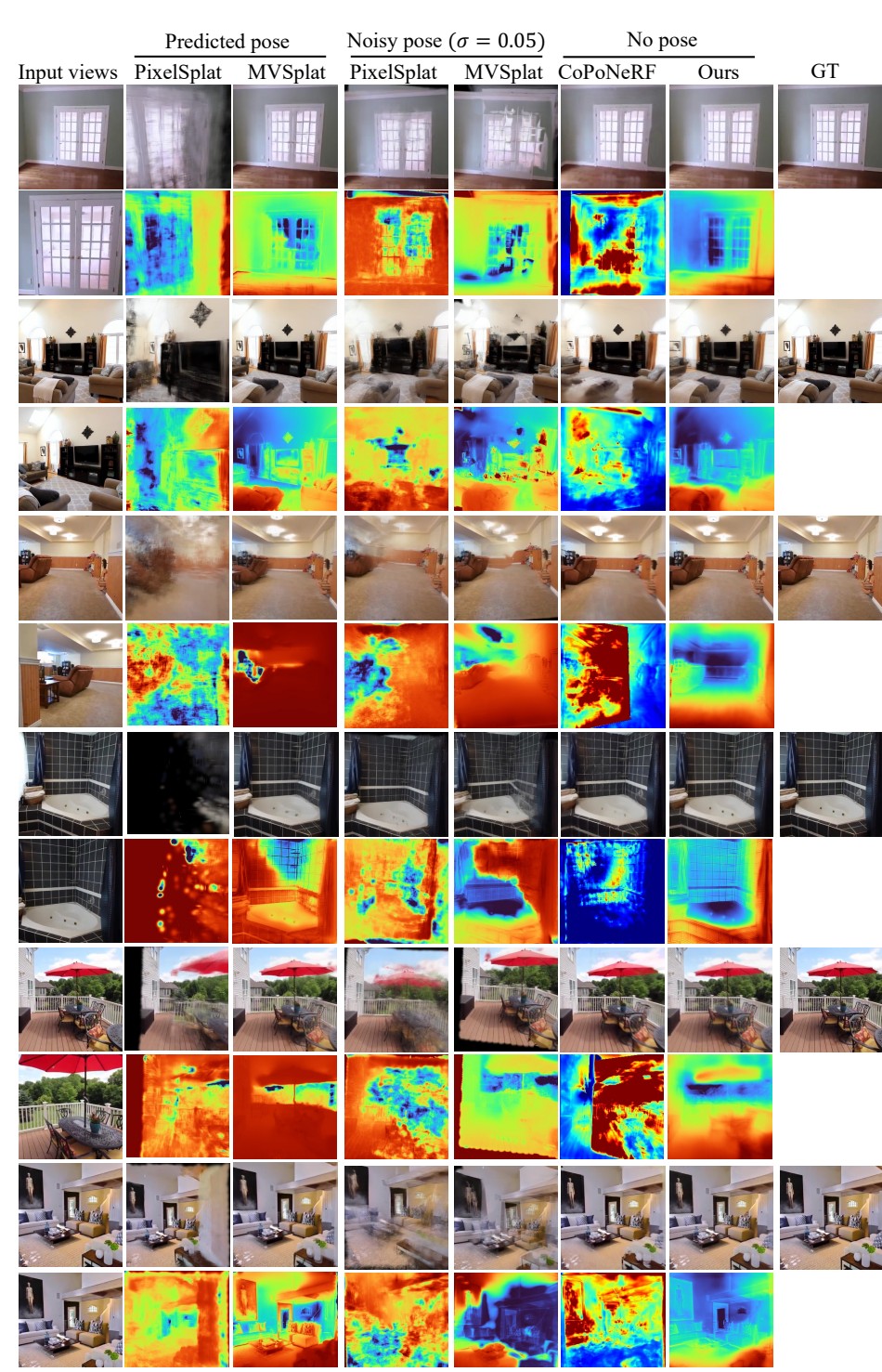

Figure 16: **Rendering and Depth comparison on RealEstate10K** The visualized images are rendered target images given 2 input views. The predicted pose indicates poses predicted using DUSt3R (Wang et al., 2024).

