# OpenReview forum: "SHARE: Bridging Shape and Ray Estimation for Pose-Free Generalizable Gaussian Splatting"
_ICLR.cc/2025/Conference — ICLR 2025 Conference Withdrawn Submission_

### Official Review · Reviewer_f2vj · 2024-10-28

**Soundness:** 3
**Presentation:** 4
**Contribution:** 3
**Rating:** 6
**Confidence:** 4

**Summary:**

This paper introduces a pose-free generalizable Gaussian Splatting framework that leverages a feed-forward network to directly regress camera poses and Gaussians from unposed sparse RGB images. The authors propose two key modules to enhance the performance: Ray-Guided Multi-View Fusion, which consolidates multi-view features into a canonical volume using Plücker rays for pose estimation and scene geometry estimation, and Anchor-Aligned Gaussian Prediction, which predicts anchor points and offsets to generate refined Gaussian Splatting for detailed reconstruction. These modules enable the proposed framework outperform previous methods on benchmarks like DTU and RealEstate10K.

**Strengths:**

- The paper is clearly written with informative illustrations. The proposed framework is intuitive and the motivation behind each module is well-explained. The evaluation and ablation results validate the impact of the proposed components.
- The idea of coarse-to-fine Gaussian splatting generation using anchor-aligned Gaussian prediction is innovative and effective.

**Weaknesses:**

- The main concern is the lack of comparison with state-of-the-art (SOTA) pose estimation methods like COLMAP, DUSt3R[1], and MASt3R[2]. The proposed method should compare baselines like camera poses from COLMAP or DUSt3R plus MVSplat/PixelSplat. While I expect COLMAP may not perform very well given the sparse image input,  DUSt3R/MASt3R is promising to give relatively accurate pose estimiation, as the paper InstantSplat[3] shows.
- The paper lacks evaluation in cross-dataset or in-the-wild settings, which raises concerns about the generalizability of the proposed methods, particularly in terms of pose estimation.
  [1]: Wang S, Leroy V, Cabon Y, et al. Dust3r: Geometric 3d vision made easy[C]//Proceedings of the IEEE/CVF Conference on Computer Vision and Pattern Recognition. 2024: 20697-20709.
  [2]: Leroy V, Cabon Y, Revaud J. Grounding Image Matching in 3D with MASt3R[J]. arXiv preprint arXiv:2406.09756, 2024.
  [3]: Fan Z, Cong W, Wen K, et al. Instantsplat: Unbounded sparse-view pose-free gaussian splatting in 40 seconds[J]. arXiv preprint arXiv:2403.20309, 2024.

**Questions:**

- It's acceptable that the method is unable to outperforms pixelsplat/ MVSplat with GT pose assumption, since it's imfeasible to obtain such accurate poses using sparse image input. But as mentioned in the weekness, we need to see whether it can outperform other methods using SOTA camera pose estimation.
- Are the baselines shown in table trained with GT cmaera poses or noisy camera poses?
- The implementation details of other baselines seem to be missing in the Appendix/Supplementary, which is cliamed in line 405. The detailed implementation of all the network structure is also missing, as cliamed in line 419.
- The equation related to \delta p is missing in equation 5. Based on Figure 4, it appears to be derived using network f_p. However, f_p in Equation 5 is used to generate Gaussian attributes, which creates some misalignment between the figure and the equation.

---

> ### Author Response · Authors · 2024-11-22
> **Response to Reviewer f2vj (1/2)**
>
> > **The main concern is the lack of comparison with state-of-the-art (SOTA) pose estimation methods like COLMAP, DUSt3R[1], and MASt3R[2].**
> >
>
> | Method | Rot. ↓ | Trans. ↓ |
> | --- | --- | --- |
> | COLMAP | 7.10 | 31.62 |
> | Relpose++ | 19.56 | 44.18 |
> | RayRegression | 3.10 | 6.57 |
> | DUSt3R | 1.77 | 13.66 |
> | MASt3R | 2.40 | 3.52 |
> | **Ours** | 2.74 | 6.28 |
>
> Regarding pose estimation, we compare our method with COLMAP[1], Relpose++[2], and RayRegression from the Cameras as Rays[3] as well as Mast3R [4] an4d Dust3R [5], which serve as foundational 3D reconstruction models for reference. While Mast3R and Dust3R demonstrate superior pose estimation performance, they rely on ground-truth dense depth maps and are trained on large-scale datasets. In contrast, our method and the other compared approaches are trained solely on the DTU train sets. We also emphasize that our primary objective is to advance pose-free reconstruction by minimizing reliance on accurate pose information.
>
> One of the key findings is that the **joint training of pose estimation and 3D Gaussians with embedded estimated poses** plays a crucial role in leveraging a multi-view geometry prior, improving quality both in rendering and pose estimation. This joint optimization process enhances the overall robustness and generalizability of our method, particularly in scenarios with limited or no ground-truth depth annotations. These results show the effectiveness of our approach in achieving accurate pose estimation as a byproduct of our pose-free rendering framework.
> We added the discussion in **Appendix** **A.4**.
>
> [1] Schonberger, Johannes L., and Jan-Michael Frahm. "Structure-from-motion revisited." CVPR, 2016
>
> [2] Lin, Amy, et al. "Relpose++: Recovering 6d poses from sparse-view observations." *3DV*, 2024.
>
> [3]Zhang, Jason Y., et al. "Cameras as Rays: Pose Estimation via Ray Diffusion.", NeurIPS, 2024.
>
> [4] Wang S, Leroy V, Cabon Y, et al. “Dust3r: Geometric 3d vision made easy” CVPR, 2024
>
> [5]: Leroy V, Cabon Y, Revaud J. “Grounding Image Matching in 3D with MASt3R”, ECCV 2024
>
> ---
>
> > **The proposed method should compare baselines like camera poses from COLMAP or DUSt3R plus MVSplat/PixelSplat. While I expect COLMAP may not perform very well given the sparse image input, DUSt3R/MASt3R is promising to give relatively accurate pose estimiation, as the paper InstantSplat[3] shows.**
> >
>
> > **It's acceptable that the method is unable to outperforms pixelsplat/ MVSplat with GT pose assumption, since it's imfeasible to obtain such accurate poses using sparse image input. But as mentioned in the weekness, we need to see whether it can outperform other methods using SOTA camera pose estimation.**
> >
>
> We appreciate the reviewer’s insightful concern and fully recognize the importance of a thorough comparison. To address this, we evaluated PixelSplat and MVSplat using state-of-the-art pose estimators, DUSt3R[1] and MASt3R[2].
>
> | Method | Pose | Rot. ↓ | Trans. ↓ | PSNR ↑ | SSIM ↑ | LPIPS ↓ |
> | --- | --- | --- | --- | --- | --- | --- |
> | **PixelSplat** | **GT** | - | - | 20.96 | 0.65 | 0.31 |
> |  | COLMAP | 7.10 | 31.62 | 13.49 | 0.34 | 0.66 |
> |  | MASt3R | 2.40 | **3.52** | 15.69 | 0.40 | 0.50 |
> |  | DUSt3R | **1.77** | 13.66 | 15.98 | 0.42 | 0.47 |
> |  | Ours | 2.74 | 6.28 | 13.29 | 0.31 | 0.66 |
> | **MVSplat** | **GT** | - | - | 21.00 | 0.69 | 0.24 |
> |  | COLMAP | 7.10 | 31.62 | 14.69 | 0.44 | 0.46 |
> |  | MASt3R | 2.40 | **3.52** | 13.31 | 0.31 | 0.58 |
> |  | DUSt3R | **1.77** | 13.66 | 13.22 | 0.32 | 0.58 |
> |  | Ours | 2.74 | 6.28 | 14.08 | 0.33 | 0.51 |
> | **Ours** |  | 2.74 | 6.28 | **19.94** | **0.63** | **0.28** |
>
> As shown in the table, our method outperforms these baseline combinations with state-of-the-art pose predictors. Additionally, as highlighted in **Table 1** and **Table 2** of our main paper, pose-dependent generalization methods are sensitive to even subtle pose errors (σ=0.01), which is a lower error margin than typical SOTA predictors achieve. We added the discussion in **Appendix** **A.2**. We hope this experiment addresses your concern, and we remain open to providing further clarifications if necessary.
>
> [1] Wang S, Leroy V, Cabon Y, et al. “Dust3r: Geometric 3d vision made easy” CVPR, 2024
>
> [2]: Leroy V, Cabon Y, Revaud J. “Grounding Image Matching in 3D with MASt3R”, ECCV 2024

---

> ### Author Response · Authors · 2024-11-22
> **Response to Reviewer f2vj (2/2)**
>
> > **The paper lacks evaluation in cross-dataset or in-the-wild settings, which raises concerns about the generalizability of the proposed methods, particularly in terms of pose estimation.**
> >
>
> **We kindly direct the reviewer to the first response of the shared official comments,** which provide comprehensive cross-dataset evaluations. We hope these comments cover the points raised and align with the context of this question.
>
> ---
>
> > **Are the baselines shown in table trained with GT cmaera poses or noisy camera poses?**
> >
>
> The baselines are trained with GT camera poses, as clarified in the updated **Appendix A.2**, where we provide detailed explanations of the baseline design. In the early stages of our research, we explored the idea of training a generalizable 3DGS model with predicted poses or in an end-to-end manner alongside a pose prediction model.
>
> However, we found that using noisy or predicted poses caused instability during training and a tendency to converge to local optima. This instability arises from the sensitivity of pixel-aligned methods (e.g., MVSplat) to pose estimation accuracy, where even small inaccuracies can lead to geometric misalignments (as conceptualized in **Fig 2**). Consequently, these methods struggle to train effectively without highly accurate poses. To address this, we have expanded on the discussion in **Appendix A.2** and provided a visualization of training failure in **Figure 10** for further clarity.
>
> ---
>
> > **The implementation details of other baselines seem to be missing in the Appendix/Supplementary, which is cliamed in line 405.**
> >
>
> > **The detailed implementation of all the network structure is also missing, as cliamed in line 419.**
> >
>
> > **The equation related to \delta p is missing in equation 5. Based on Figure 4, it appears to be derived using network f_p. However, f_p in Equation 5 is used to generate Gaussian attributes, which creates some misalignment between the figure and the equation.**
> >
>
> We sincerely thank the reviewer for pointing out these important details. We apologize for the oversight in the earlier version and have now included detailed implementation descriptions in **Appendix A.2. and A.3.**. Additionally, the inconsistencies between **Equation 5** and **Figure 4** have been addressed and revised in both the text and the figure for improved clarity.

---

> > ### Comment · Reviewer_f2vj · 2024-11-25
> > **Response to the authors**
> >
> > Thanks for the detailed response and the additional experiments. Could you clarify why you mentioned that *Mast3R* and *Dust3R* rely on ground-truth dense depth maps? From my understanding, their methods are capable of obtaining camera poses solely from RGB inputs.
> > Additionally, I understand that training *PixelSplat* or *MVSplat* with added noises may result in unstable training. However, is it possible to train them using the estimated poses from *MASt3R* or *DUSt3R*? I assume the last table you provided uses their poses only during inference, which might cause the different pose distribution from training pose distribution. My other concerns are resolved.

---

> > > ### Author Response · Authors · 2024-11-26
> > > **Response to Reviewer f2vj**
> > >
> > > Dear reviewer f2vj,
> > >
> > > Thank you for your response and for bringing up the detailed discussion.
> > >
> > > > Could you clarify why you mentioned that Mast3R and Dust3R rely on ground-truth dense depth maps?
> > >
> > > As you mentioned, Dust3R and MASt3R only require RGB inputs during inference. What we wanted to clarify was that these methods use ground-truth depth maps during training, whereas our approach does not require ground-truth depth maps at any stage of training.
> > >
> > > ---
> > >
> > > > However, is it possible to train them using the estimated poses from MASt3R or DUSt3R?
> > >
> > > We sincerely appreciate your detailed discussion and agree that this is an interesting experiment. To verify this, we trained MVSplat using poses predicted with dust3r. However, even with the poses predicted with DUSt3R[1], we found that the training of such baselines remains unstable and leads to degenerate performances. As we have claimed, this instability arises from the high sensitivity of these baselines to pose errors. To further clarify this discussion, we have updated the visualization in **Figure 10** in **Appendix A.2**.
> > >
> > > [1] Wang S, Leroy V, Cabon Y, et al. “Dust3r: Geometric 3d vision made easy” CVPR, 2024

---

> ### Author Response · Authors · 2024-11-28
> **Official comment for Reviewer f2vj**
>
> As the PDF revision period period is drawing to a close and the discussion deadline is approaching, we would like to kindly remind you of the improvements we made to our paper, thanks to your valuable feedback.
>
> - We compared our method's **pose estimation performance** with various pose estimators. This comparison with RayRegression[1], which only learns pose, showed that jointly learning shape with pose shows a synergetic effect, improving pose estimation performance.
> - We combined pose-dependent baselines (pixelSplat[2], MVSplat[3]) with various state-of-the-art pose estimators, enhancing the **thoroughness of the evaluation and baselines**. This experiment further supports our claim that simply combining pose estimators with pose-dependent methods leads to geometry misalignment, which in turn results in degraded view synthesis performance.
> - We conducted additional experiments on the ACID[4] and BlendedMVS[5] datasets, demonstrating the **cross-dataset generalizability** of our method.
> - We included results indicating that **training pose-required methods with estimated poses can lead to instability**, which highlights the importance of our proposed approach.
> - We made corrections to certain details in the paper, which helped improve its overall presentation.
>
> We sincerely appreciate your careful review, and we’re glad we could address your concerns. We hope these changes and experimental improvements will be helpful as you finalize your review.
>
> ---
>
> [1] Zhang, Jason Y., et al. "Cameras as Rays: Pose Estimation via Ray Diffusion.", NeurIPS, 2024.
>
> [2] Charatan, David, et al. "pixelsplat: 3d gaussian splats from image pairs for scalable generalizable 3d reconstruction." CVPR, 2024.
>
> [3] Chen, Yuedong, et al. "Mvsplat: Efficient 3d gaussian splatting from sparse multi-view images." ECCV, 2024.
>
> [4] Liu, Andrew, et al. "Infinite nature: Perpetual view generation of natural scenes from a single image." ICCV. 2021.
>
> [5] Yao, Yao, et al. "Blendedmvs: A large-scale dataset for generalized multi-view stereo networks." CVPR. 2020.

---

> > ### Comment · Reviewer_f2vj · 2024-11-30
> > **Response to the authors**
> >
> > Thank you for your efforts throughout the rebuttal. All my concerns have been resolved.

---

> > > ### Author Response · Authors · 2024-12-01
> > > **Official comment for Reviewer f2vj**
> > >
> > > I'm glad to hear that the concerns have been addressed. If you have any additional concerns, please feel free to share them, and I'll be happy to address them promptly.

---

### Official Review · Reviewer_RF3w · 2024-10-30

**Soundness:** 3
**Presentation:** 2
**Contribution:** 3
**Rating:** 6
**Confidence:** 3

**Summary:**

In this work, the authors predict a framework for pose-free generaliable 3DGS primitives prediction. By jointly predicting camera poses described with Plucker rays and injecting them into the Canonical Volume Construction, the framework can integrate multi-view features into geometry volume and feature volume under a single canonical view, which would be used for subsequent Gaussian primitive prediction through MLPs.

**Strengths:**

1. The idea of introducing  Plucker ray for pose representations and multi-view fusion guidance is useful;
 2. The Anchor-aligned Gaussian prediction by integrating cost volumes from multiple views into a single canonical view is a interesting idea;
 3. The evaluations on DTU and RealEstate10K datasets confirm that the proposed method can predict higher quality 3DGS primitives under the pose-free setting;

**Weaknesses:**

The main weaknesses of this work may include the lack of some critical details and discussions about related works. Please check the questions section for my problems about the details.
As for the lack of discussions, the idea of introducing Plucker ray maps to represent camera poses has been introduced in CAT3D[1]. The authors should discuss about their differences, at least.
As for the pose-free generalizable prediction of 3DGS primitives, Splat3R[2] also proposes another effective solution by estimating the camera poses through DUST3R[3]. Some related discussions and comparisons should be necessary to validate the effectiveness of this work.
[1] Cat3d: Create anything in 3d with multi-view diffusion models
[2] Splatt3r: Zero-shot gaussian splatting from uncalibarated image pairs
[3] Dust3r: Geometric 3d vision made easy

**Questions:**

1. How is the cost volume $C_i$ transformed into pose-aware cost-volume $C_i'$? Cost volumes directly calculated from different views should not be directly added. Is there any operation, such as alignment, to make them additive? I would also appreciate it if the authors can provide more details about the $V_f$ and Fig.4.
 2. The necessity of using Plucker rays for camera poses should be further confirmed through some ablation experiments. For example, can we directly regress the camera poses with GT poses?
 3. How is the ground truth  Plucker rays acquired? Are they acquired with the original camera poses?
 4. How is the performances on more challenging dataset, e.g., Scannet? Besides, I am also curious about the number of Gaussian primitives acquired by this framework. As only one canonical view is used for prediction, would the number of predicted primitives less than other methods?

**Details Of Ethics Concerns:**

NA.

---

> ### Author Response · Authors · 2024-11-22
> **Response to Reviewer RF3w (1/3)**
>
> > As for the lack of discussions, the idea of introducing Plucker ray maps to represent camera poses has been introduced in CAT3D[1]. The authors should discuss about their differences, at least. Some related discussions and comparisons should be necessary to validate the effectiveness of this work.
> >
>
> We sincerely thank the reviewer for the insightful suggestion, which has significantly enhanced the clarity and depth of our work. In response, we have incorporated a discussion on CAT3D [1] and related methods that utilize Plücker rays as pose information in **Section 3** of our paper.
>
> To clarify the distinction, our method fundamentally differs from these approaches by operating in a **pose-free setting**. While methods like CAT3D leverage ray-represented ground-truth poses as conditioning inputs or feature embeddings, our approach jointly learns to predict Plücker rays and 3D Gaussians in a feed-forward manner directly from input images. The predicted rays are integral to our multi-view fusion pipeline, as they provide geometric guidance to resolve ambiguities inherent in pose-free scenarios. This design enables our method to achieve robust performance without relying on explicit pose information, setting it apart from prior works.
>
> [1] Cat3d: Create anything in 3d with multi-view diffusion models
>
> ---
>
> > As for the pose-free generalizable prediction of 3DGS primitives, Splat3R[2] also proposes another effective solution by estimating the camera poses through DUST3R[3].
> >
>
> Thank you for your valuable suggestions. We have compared our work with the concurrent work Splatt3R[1], which utilizes pre-trained MASt3R[2] weights for geometry estimation. We observed that Splatt3R faces a significant scale-ambiguity issue when applied to out-of-distribution data not seen during the training of MASt3R. The estimated scale of the reconstructed point clouds often misaligns with the scale of the camera poses for novel view rendering.
>
> To address this, we attempted to fine-tune Splatt3R on the target datasets (RealEstate10K and DTU) using photometric loss. However, this approach led to convergence issues, with the model output blurry reconstructions. This behavior can be attributed to Splatt3R's reliance on geometry estimation from MASt3R, which requires ground-truth dense depths to mitigate the scale-ambiguity issue. Unfortuantely, our target datasets present challenges in this regard: RealEstate10K lacks ground-truth depths, and DTU provides only sparse, masked depth maps, making it difficult to adapt Splatt3R directly without significant modifications.
>
> To provide a fair baseline, we evaluated the pre-trained Splatt3R model (trained on ScanNet++) directly on our datasets under its original training conditions. We included both in-dataset (Table A and B) and cross-dataset (Table C) generalization tests. We have included these results in the supplementary material, **Appendix A.4**, with a detailed discussion of the experimental settings, evaluation metrics, and qualitative visualizations.
>
> - Table A (comparison with Splatt3R on DTU dataset)
> | DTU | PSNR ↑ | SSIM ↑ | LPIPS ↓ |
> |---|---|---|---|
> | Splatt3R | 11.78 | 0.28 | 0.57 |
> | Ours | **17.50** | **0.34** | **0.48** |
> - Table B (comparison with Splatt3R on RealEstate10K dataset)
> | RealEstate10K | PSNR ↑ | SSIM ↑ | LPIPS ↓ |
> |---|---|---|---|
> | Splatt3R | 15.80 | 0.53 | 0.30 |
> | Ours | **21.23** | **0.71** | **0.26** |
> - Table C (comparison with Splatt3R on cross-dataset generalization test with ACID[3] datasets)
> | **ACID** | **Training Data** | PSNR ↑ | SSIM ↑ | LPIPS ↓ |
> |---|---|---|--|---|
> | Splatt3R | ScanNet++ | 17.49 | 0.63 | **0.26** |
> | Ours | RealEstate10K | **23.47** | **0.69** | **0.26** |
>
> Table A and B shows that our method significantly outperform on both datasets with a large margin. In addition, to discuss about the cross-dataset generalization quality, we tested our method (trained on RealEstate10K) and Splatt3R (trained on ScanNet) on the ACID[3] dataset. The table shows that our method shows superior performance in all metrics, underscoring the robustness and generalizability of our approach. These results validate our method’s effectiveness in pose-free multi-view reconstruction, even in challenging scenarios without ground-truth depth supervision.
>
> [1] Smart, Brandon, et al. "Splatt3r: Zero-shot gaussian splatting from uncalibrated image pairs." arXiv. 2024.
>
> [2] Leroy, Vincent, Yohann Cabon, and Jérôme Revaud. "Grounding image matching in 3d with mast3r." ECCV, 2024.
>
> [3] Liu, Andrew, et al. "Infinite nature: Perpetual view generation of natural scenes from a single image." ICCV. 2021.

---

> > ### Comment · Reviewer_RF3w · 2024-11-25
> > **Response to the authors**
> >
> > Thanks for the careful response of the authors. Most of my concerns are resolved. I will keep my rating.

---

> > > ### Author Response · Authors · 2024-11-26
> > >
> > > Dear Reviewer RF3w,
> > >
> > > Thank you for your thoughtful consideration and detailed discussion.
> > > Your feedback has been invaluable in helping us refine our contributions throughout the discussion process.
> > >
> > > We believe our work makes significant advancements in pose-free 3D scene modeling by effectively leveraging multi-view information without relying on additional geometric prior.
> > >
> > > If you have any further questions, please let us know.
> > > We would be happy to provide additional clarifications or results.
> > >
> > > Best regards,
> > >
> > > Authors

---

> ### Author Response · Authors · 2024-11-22
> **Response to Reviewer RF3w (2/3)**
>
> > **How is the cost volume $C_i$ transformed into pose-aware cost-volume $C_i’$? Cost volumes directly calculated from different views should not be directly added. Is there any operation, such as alignment, to make them additive?**
> >
>
> To address the additivity of cost volumes $\{C_i\}$ from different views, we perform a refinement step through cost aggregation, which is conditioned on predicted Plücker rays. Specifically, we employ a transformer-based 2D U-Net augmented with cross-attention layers, where the predicted rays serve as key-value pairs and the cost volumes act as queries. This mechanism embeds pose awareness into the cost volumes by utilizing the geometric guidance provided by the rays.
>
> ---
>
> > **I would also appreciate it if the authors can provide more details about the $V_f$ and Fig.4.**
> >
>
> We sincerely thank the reviewer for the suggestion regarding the unclear aspects of our method. We construct the global canonical volume $V_g$ as described in **Equation 3** of the paper. $V_g$ is used to estimate the anchor points, which represent a coarse structure downscaled by a factor of 4 relative to the original image resolution. Simultaneously, we build the feature volume $V_f$ in the same manner, but with upscaled features, to estimate the offset vectors and Gaussian parameters for fine detailed reconstruction. We hope this explanation addresses your concerns.
>
> Additionally, we revised and clarified the relevant sections of the paper including **Fig.4** and **Section 4.3** to ensure a clearer understanding for all readers. Thank you for bringing this to our attention.
>
> ---
>
> > **The necessity of using Plucker rays for camera poses should be further confirmed through some ablation experiments. For example, can we directly regress the camera poses with GT poses?**
> >
>
> The Plücker ray representation is a fundamental component of our pipeline, as it enables seamless integration of camera poses into the multi-view feature aggregation process. While using a 6D pose representation could lead to an alternative option, our method builds upon the established assumption from Cameras as Rays [1] that ray-based representations offer advantages for learning. Specifically, Cameras as Rays reports improved stability and accuracy when using ray-based methods, as evidenced by the comparison of "R+T Regression" and "Ray Regression" in Tables 1 and 2 of their paper.
>
> Additionally, we would like to respectfully note that directly regressing 6D poses would not constitute a fair ablation of the ray-based representation, as it would involve not only altering the pose representation but also redesigning the embedding strategy.
>
> [1] Zhang, Jason Y., et al. "Cameras as Rays: Pose Estimation via Ray Diffusion.", NeurIPS, 2024.

---

> ### Author Response · Authors · 2024-11-22
> **Response to Reviewer RF3w (3/3)**
>
> > **How is the ground truth Plucker rays acquired? Are they acquired with the original camera poses?**
> >
>
> The ground truth Plücker rays are derived from the conventional 4 $\times$ 4 extrinsic matrices available during training. These matrices are directly transformable to Plücker ray representations and vice versa, as described in Section 4.1 of our paper. For this transformation, we adhere to the formulation detailed in Cameras as Rays [1].
>
> It is important to emphasize that this reliance on camera poses is strictly limited to the training phase. During inference, our method completely eliminates the need for any explicit pose assumptions, ensuring a fully pose-independent inference pipeline.
>
> [1] Zhang, Jason Y., et al. "Cameras as Rays: Pose Estimation via Ray Diffusion.", NeurIPS, 2024.
>
> ---
>
> > **How is the performances on more challenging dataset, e.g., Scannet?**
> >
>
> To evaluate our method across datasets of varying scales, we conducted experiments on the DTU dataset, which contains smaller scenes, and the RealEstate10K dataset, which includes larger indoor and outdoor scenes. Additionally, we rigorously assessed the cross-dataset generalization capabilities of our approach on the ACID [1] and BlendedMVS [2] datasets, both of which feature diverse indoor and outdoor scenarios, to further demonstrate the robustness of our method. We kindly direct the reviewer to the common comments in the top-level response.
>
> We appreciate the reviewers’ suggestion to test on additional datasets, such as ScanNet[3] or ScanNet++[4], to further validate our approach. While obtaining access to large-scale datasets requires additional time due to permission and resource constraints, we plan to include results on such datasets and ensure they are incorporated into the final revision. This extension will provide a more comprehensive evaluation of our method.
>
> [1] Liu, Andrew, et al. "Infinite nature: Perpetual view generation of natural scenes from a single image." *ICCV. 2021.
>
> [2]  Yao, Yao, et al. "Blendedmvs: A large-scale dataset for generalized multi-view stereo networks." CVPR. 2020.
>
> [3] Dai et al. "ScanNet: Richly-annotated 3D Reconstructions of Indoor Scenes" CVPR. 2017.
>
> [4] Yeshwanth et al. "ScanNet++: A High-Fidelity Dataset of 3D Indoor Scenes" ICCV. 2023
>
> ---
>
> > **Besides, I am also curious about the number of Gaussian primitives acquired by this framework. As only one canonical view is used for prediction, would the number of predicted primitives less than other methods?**
> >
>
> Our approach is designed to maintain a fixed number of Gaussian primitives, regardless of the number of input views. This is achieved by predicting 3 offsets per Gaussian, as detailed in the main paper. This strategy ensures an efficient and consistent representation while effectively capturing the scene geometry.
>
> In contrast, baseline methods typically experience a linear increase in the number of Gaussians as the number of input views grows, resulting in higher computational costs. Furthermore, our design is inherently robust to geometry misalignments caused by pose errors, as it avoids introducing any misalignment in 3D space, ensuring stable representations.

---

### Official Review · Reviewer_QjrW · 2024-11-01

**Soundness:** 2
**Presentation:** 2
**Contribution:** 2
**Rating:** 5
**Confidence:** 4

**Summary:**

This paper focuses on generalizable Gaussian splatting from sparse unposed images. To this end, it employs the Pl$\ddot{u}$cker ray representation for relative pose estimation. Based on the ray representation, it builds cost volumes from extracted image features. Moreover, it embeds the ray representation into the cost volumes using patch-wise cross attention. After aggregating these cost volumes, a geometry volume and feature volume are obtained to construct Gaussians. This works employs anchor points to distribute local Gaussians. By optimizing both the Gaussians and ray representation, it can recover the pose and 3D scene at the same time. Experiments on DTU and RealEstate10K verify the effectiveness of the proposed method.

**Strengths:**

1. This paper introduces the Pl$\ddot{u}$cker ray representation for relative pose estimation instead of directly predicting camera rotation and translation.
2. Based on the pose representation, the proposed method embeds learned pose information into the cost volume to improve the Gaussian learning.
3. For Gaussian learning, this work leverage anchor points to distribute local Gaussian, which can hierarchically learn intricate textures or complex geometries.

**Weaknesses:**

1. The proposed method relies on cost volume construction, which requires depth range priors. Moreover, can you discuss the limitation of the proposed unable on tackling unbounded $360^{\circ}$ scenes?
2. In fact, the anchor point idea used in this work is proposed by Scaffold-GS [1].  Can you clarify the difference between your use and Scaffold-GS? Maybe it is better to present a preliminary to introduce it as the scene representation.
[1]  Lu, Tao, et al. Scaffold-gs: Structured 3d gaussians for view-adaptive rendering. Proceedings of the IEEE/CVF Conference on Computer Vision and Pattern Recognition. 2024.
3. The proposed method are trained on DTU and RealEstate10K, respectively. Then, the trained models are used to test the corresponding datasets. This cannot verify the generalizable ability of the proposed method. Can you test the proposed method on RealEstate10K with the model trained on DTU, and test the proposed method on DTU with the model trained on RealEstate10K?

**Questions:**

1. For the generalizable ability, can the model trained on one dataset generalize to different datasets? For example, can the model trained on RealEstate10K be used to test Tanks and Temples datasets [2]?
[2] Knapitsch, Arno, et al. "Tanks and temples: Benchmarking large-scale scene reconstruction." ACM Transactions on Graphics (ToG) 36.4 (2017): 1-13.
2. This work uses the ray representation from RayDiffusion. Can you compare the pose estimation performance with RayDiffusion?
3. For the pose-required methods, such as MVSplat, I am wondering if their rendering performance will improve if they are trained with the pose information estimated by  the proposed method or RayDiffusion.
4. In fact, the weighted cost volume in Eq. (3) can reflect the complex visibility information better. Why the mean and variance-based volume is added? Can you have an experiment on this?
5. Can you show the efficiency of the proposed method in terms of inference time and GPU memory usage? Can the proposed method tackle higher-resolution input images, such as the original-resolution images in DTU and Tanks and Temples?

---

> ### Author Response · Authors · 2024-11-22
> **Response to Reviewer QjrW (1/3)**
>
> > **The proposed method relies on cost volume construction, which requires depth range priors.**
> >
>
> We appreciate the reviewer’s insightful comment. While having accurate depth range priors can indeed enhance reconstruction quality, our approach does not necessarily rely on strict depth range constraints. For instance, in the RealEstate10K dataset, where ground-truth depth ranges are unavailable, we employed a broad range of 1 to 100, demonstrating the robustness of our method to varying scales. This flexibility underscores the generalizability of our framework across various scenes without the need for precise prior depth range information.
>
> ---
>
> > **Moreover, can you discuss the limitation of the proposed unable on tackling unbounded 360∘ scenes?**
> >
>
> We fully recognize the growing need and demand for addressing more complex scenarios, such as 360-degree input images. Recent advancements, such as MVSplat360 [1], have made significant progress in tackling these challenges. We believe that our method can be integrated with such approaches, offering the potential to further enhance solutions for these demanding cases. We appreciate your suggestion and will include more detailed discussions on these limitations in the revised manuscript.
>
> [1] Chen, Yuedong, et al. "MVSplat360: Feed-Forward 360 Scene Synthesis from Sparse Views." NeurIPS, 2024
>
> ---
>
> > **In fact, the anchor point idea used in this work is proposed by Scaffold-GS [1]. Can you clarify the difference between your use and Scaffold-GS? Maybe it is better to present a preliminary to introduce it as the scene representation.**
> >
>
> We sincerely thank the reviewer for bringing up Scaffold-GS [1] and appreciate the opportunity to clarify the distinctions between our approach and theirs. While both methods utilize anchor points, their objectives and feature characteristics differ fundamentally.
>
> In Scaffold-GS, anchor points are voxelized centers derived from SfM reconstructions, designed to enhance local fidelity by constraining Gaussian primitives to localized offsets. This process relies on iterative, per-scene optimization to align with pseudo ground-truth structures, focusing on improving local accuracy for specific scenes.
>
> In contrast, our method predicts pixel-aligned anchor points and their corresponding features in a feed-forward, data-driven manner, bridging 2D image information to 3D scene representation. Unlike Scaffold-GS, our approach generalizes to unseen scenes without iterative optimization and models Gaussian primitives that capture global scene structures, supporting multiple views effectively.
>
> These distinctions underline the fundamental differences in anchor point usage and methodology. If there’s any additional concerns regarding this, we are glad to further discuss on this topic.
>
> [1] Lu, Tao, et al. Scaffold-gs: Structured 3d gaussians for view-adaptive rendering. CVPR, 2024.
>
> ---
>
> > **The proposed method are trained on DTU and RealEstate10K, respectively. Then, the trained models are used to test the corresponding datasets. This cannot verify the generalizable ability of the proposed method. Can you test the proposed method on RealEstate10K with the model trained on DTU, and test the proposed method on DTU with the model trained on RealEstate10K?**
> >
>
> > **For the generalizable ability, can the model trained on one dataset generalize to different datasets? For example, can the model trained on RealEstate10K be used to test Tanks and Temples datasets [2]?**
> >
>
> We have addressed this cross-dataset generalization evaluation in the shared official comment. As in the comment, we evaluated SHARE trained on RealEstate10K and tested it on ACID. In addition, we tested cross-dataset generalization from DTU to BlendedMVS datasets. Since the difference distribution and the number of scenes between DTU ( < 100 scenes of object-centric data with black background) and RealEstate10K (60K+ large-scale of indoor and outdoor scenes). We hope this provides clarity, and we are happy to address any further questions or concerns.

---

> ### Author Response · Authors · 2024-11-22
> **Response to Reviewer QjrW (2/3)**
>
> > **This work uses the ray representation from RayDiffusion. Can you compare the pose estimation performance with RayDiffusion?**
>
> | Method | Rot. ↓ | Trans. ↓ |
> | --- | --- | --- |
> | COLMAP | 7.10 | 31.62 |
> | Relpose++ | 19.56 | 44.18 |
> | RayRegression | 3.10 | 6.57 |
> | DUSt3R | 1.77 | 13.66 |
> | MASt3R | 2.40 | 3.52 |
> | **Ours** | 2.74 | 6.28 |
>
> We compared the performance of our pose estimation method against state-of-the-art approaches, including **RayRegression** proposed in the RayDiffusion paper [1]. We would like to respectfully clarify that we chose to compare with RayRegression rather than RayDiffusion because, as noted in **Table 6** of the RayDiffusion [1] appendix, regression-based methods require only 0.1 seconds for inference, whereas diffusion-based methods (RayDiffusion) take 11.1 seconds. While diffusion-based approaches may offer slightly improved performance, their slower inference times make them less suitable for our pipeline, which prioritizes efficiency.
>
> Our method demonstrates improved performance compared to RayRegression, highlighting the synergistic benefits of jointly optimizing shape and camera rays. Meanwhile, it is important to emphasize that our method is primarily designed for pose-free novel view rendering, with pose estimation being an auxiliary outcome of the process.
>
> [1] Zhang, Jason Y., et al. "Cameras as Rays: Pose Estimation via Ray Diffusion.", NeurIPS, 2024.
>
> ---
>
> > **For the pose-required methods, such as MVSplat, I am wondering if their rendering performance will improve if they are trained with the pose information estimated by the proposed method or RayDiffusion.**
> >
>
> In the early stages of our research, we explored the idea of training a generalizable 3DGS model with predicted poses or in an end-to-end manner alongside a pose prediction model. However, we found that using noisy or predicted poses caused instability during training and a tendency to converge to local optima.
> This instability arises from the sensitivity of pixel-aligned methods (e.g., MVSplat) to pose estimation accuracy, where even small inaccuracies can lead to geometric misalignments (as conceptualized in **Figure 2**). Consequently, these methods struggle to train effectively without highly accurate poses. We have expanded on the discussion in **Appendix A.2** and provided a visualization of training failure in **Figure 10** for further clarity.
>
> ---
>
> > **In fact, the weighted cost volume in Eq. (3) can reflect the complex visibility information better. Why the mean and variance-based volume is added? Can you have an experiment on this?**
> >
>
> We appreciate the reviewer’s suggestion for this analysis. As explained in **Section 4.2** on canonical volume construction, the mean-variance volume is designed to mitigate the risk of a trivial solution where a single view disproportionately dominates the fusion process, drawing inspiration from MVSNet [1]. This design enhances training stability by ensuring balanced contributions from all views, avoiding over-reliance on any single view. Our experimental results demonstrate improved performance with the inclusion of mean-variance volume.
>
> | Method | PSNR ↑ | SSIM ↑ |  LPIPS ↓ |
> | --- | --- | --- | --- |
> | w/o mean-var volume | 18.10 | 0.55 | 0.33 |
> | Ours | **19.09** | **0.64** | **0.29** |
>
> [1] Yao, et al. "Mvsnet: Depth inference for unstructured multi-view stereo." *ECCV,* 2018.
>
> ---
>
> > **Can you show the efficiency of the proposed method in terms of inference time and GPU memory usage?**
> >
>
> We thank the reviewer for their valuable suggestion. Following the recommendation, we measured the inference time efficiency and GPU memory consumption of our method, alongside CoPoNeRF[1], MVSplat combined with pose estimator (MASt3R[2]), and Splatt3R[3] on the RealEstate10K dataset. We tested all experiments with RTX 3080. The results are presented in the table below.
>
> | Method | Inference Time (s) | GPU Memory (MB) |
> | --- | --- | --- |
> | CoPoNeRF | 3.37 | 9587.22 |
> | MVSplat + Mast3r | 0.22 | **4376.94** |
> | Splatt3R | 0.26 | 6198.00 |
> | **Ours** | **0.17** | 5887.18 |
>
> We would also like to highlight that our approach achieves superior rendering quality compared to all baselines. Additional experimental details and the corresponding table have been included in **Appendix A.4** for further reference.
>
> [1] Hong, Sunghwan, et al. "Unifying Correspondence Pose and NeRF for Generalized Pose-Free Novel View Synthesis." CVPR, 2024
>
> [2] Leroy, Vincent, Yohann Cabon, and Jérôme Revaud. "Grounding image matching in 3d with mast3r." *ECCV*, 2025.
>
> [3] Smart, Brandon, et al. "Splatt3r: Zero-shot gaussian splatting from uncalibrated image pairs." arXiv. 2024.

---

> ### Author Response · Authors · 2024-11-22
> **Response to Reviewer QjrW (3/3)**
>
> > **Can the proposed method tackle higher-resolution input images, such as the original-resolution images in DTU and Tanks and Temples?**
> >
>
> Our method does not impose explicit restrictions on image resolution. However, as we adopt a pixel-aligned Gaussian prediction approach, higher-resolution images inherently result in a larger number of Gaussian primitives, which can lead to increased memory consumption. This characteristic is not unique to our method but is a common limitation of pixel-aligned approaches, including our baselines (PixelSplat, MVSplat).
>
> However, a key distinction of our method is that the number of Gaussians remains fixed regardless of the number of input views, as our method only predicts the Gaussians from the canonical view with the fused features. This suggests that, for high-resolution reconstructions requiring multiple views, our approach handles the task more efficiently with less Gaussians required compared to the baselines.

---

> ### Author Response · Authors · 2024-11-26
> **Kind reminder for reviewer-author discussion**
>
> Dear Reviewer QjrW,
>
> As the discussion period is ending soon, we wanted to kindly remind you of our responses to your comments.
> We truly value your feedback and are happy to answer any remaining questions or concerns you might have.
>
> Please feel free to let us know if there is any more information we can provide to help with the discussion.
> Thank you again for your time and thoughtful review.
>
> Best regards,
>
> Authors

---

> ### Author Response · Authors · 2024-11-28
> **Official Comment for Reviewer QjrW**
>
> As the PDF revision period is drawing to a close and the discussion deadline is approaching, we would like to remind you of our improvements kindly. During the review period, we were able to improve our paper thanks to your valuable feedback. Based on your suggestions, we made the following enhancements:
>
> - We clarified the differentiation of our work from additional related works, such as **Scaffold-GS[1]**.
> - We validated our method's **cross-dataset generalizability**, showing robustness on different datasets.
> - By **comparing pose estimation performance with RayRegression[2]**, we illustrated that jointly learning both shape and pose leads to a synergistic improvement in performance.
> - We included results that show how **training pose-required methods with estimated poses leads to instability**, which further emphasizes the effect of jointly learning the shape and pose.
> - Through additional **ablation studies on the mean-variance volume**, we were able to assess the effectiveness of our pipeline design further.
> - We presented an **efficiency test** that shows that our method outperforms our concurrent work, Splatt3R[3], in terms of reconstruction as well as inference time and GPU memory usage.
>
> Once again, we sincerely appreciate your valuable feedback. We would greatly appreciate it if you could take these changes and the resulting experimental improvements into consideration when finalizing your review. If you have any further questions or concerns, we are happy to address them.
>
> ---
>
> [1] Lu, Tao, et al. Scaffold-gs: Structured 3d gaussians for view-adaptive rendering. CVPR, 2024.
>
> [2] Zhang, Jason Y., et al. "Cameras as Rays: Pose Estimation via Ray Diffusion.", NeurIPS, 2024.
>
> [3] Smart, Brandon, et al. "Splatt3r: Zero-shot gaussian splatting from uncalibrated image pairs." arXiv. 2024.

---

> > ### Comment · Reviewer_QjrW · 2024-12-02
> >
> > Thanks for your response! Most of my concerns have been adressed. I still have some questions.
> > 1. I still think the 3D respresentation of this work is built upon Scaffold-GS, this representation improves the performance a lot (Table 3). Therefore, the perfomance improvement of this work may be attributed to the Scaffold-GS representation. I know this work also made some modifications, however, it is better to introduce Scaffold-GS as a preliminary first.
> >
> > 2. For W3, I cannot find cross-dataset experimental results. In addition, when generalizing the two trained models to other datasets,  how to choose one of them to test ACID or BlendedMVS?
> >
> > 3. For the noisy poses, it is better to compare with SPARF, which uses noisy poses to train NeRF and provides an effiective way to alleviate the pose noises.
> >
> > 4. For the ablation for the mean-var volume, for the ours result (PSNR: 19.09, SSIM: 0.64, LPIPS: 0.29), I cannot find correspondint resutls in the main text. What is the experimental setting for this ablation?

---

> ### Author Response · Authors · 2024-12-03
> **Official Comment for Reviewer QjrW**
>
> **Response to Q1**
>
> Thank you for your thoughtful comments. We appreciate the suggestion to introduce Scaffold-GS as a preliminary, and we agree that this will provide important context for our work. In the revised manuscript, **we will include a detailed discussion of Scaffold-GS** in **Section 4.3** of our paper, which will establish it as a foundational approach and help readers better understand how our method builds upon it. We will also **clearly highlight the two key differences** between our approach and Scaffold-GS:
>
> - Specifically, while Scaffold-GS uses voxelized centers derived from SfM reconstructions with dense views to define anchor points, our method takes a pose-free approach and predicts pixel-aligned anchor points in a canonical space in a data-driven manner.
> - Furthermore, unlike Scaffold-GS, which relies on iterative optimization and ground-truth camera poses for scene-specific adjustments, our approach generalizes to unseen scenes without requiring such optimization, providing a significant distinction in flexibility and applicability.
>
> We hope this addresses your concerns, and we are grateful for the constructive feedback.
>
> ---
>
> **Response to Q2**
>
> We apologize for any confusion regarding the cross-dataset experimental results. As you correctly pointed out, these results are mentioned in the **common comments section**, but we’ve moved the reply below for your convenience.
>
> We conducted cross-dataset experiments, evaluating our model trained with RealEstate10K dataset on ACID dataset (following pixelsplat, mvsplat), and  our model trained with DTU on BlendedMVS dataset (following sparseneus[1], uforecon[2]), following related papers.
>
> |  |  | **RealEstate10K → ACID** |  |  | **DTU → BlendedMVS** |  |  |
> |-|-|-|-|-|-|-|-|
> |Method|Pose|**PSNR↑**|**SSIM↑**|**LPIPS↓**|**PSNR↑**|**SSIM↑**|**LPIPS↓**|
> |**PixelSplat** | GT | 26.84 | 0.81 | 0.18 | 11.64 | 0.20 | 0.67 |
> |  |σ = 0.01 | 21.73 | 0.57 | 0.28 | 11.65 | 0.20 | 0.68 |
> |**MVSplat**|GT|28.18|0.84|0.15|12.04 | 0.19 | 0.56 |
> |  | σ = 0.01|21.65|0.57|0.27|11.92|0.20|0.59 |
> |**Ours**|-|23.47|0.69|0.26|12.19|0.26|0.61|
>
> As shown in the table, our method exhibits strong generalizability, performing comparably to or even surpassing the baselines that utilize GT poses. We also compared the baseline methods with minimal gaussian noise level (sigma=0.01), where rotation and translation angular errors are far lower than the state-of-the-art pose estimators. We included the comprehensive quantitative (**Table 6**) and qualitative results (**Figure 11**) in the **Appendix A.4.**
>
> In choosing the evaluation datasets, we considered two factors: 1) to ensure a fair comparison, we followed the baseline methods and built upon the established conventions, and 2) the intended use of the datasets, which differ in terms of scene types (e.g., RealEstate10K → ACID for indoor/outdoor scenes) and focus (e.g., DTU → BlendedMVS for object-centered evaluation).
>
> [1] Long et al. Sparseneus: Fast generalizable neural surface reconstruction from sparse views, ECCV 2022
>
> [2] Na et al. UFORecon: Generalizable Sparse-View Surface Reconstruction from Arbitrary and UnFavOrable Data Sets, CVPR 2024
>
> ---
>
> **Response to Q3**
>
> Thank you for your suggestion. We agree that per-scene optimization methods, such as SPARF, effectively alleviate pose noise by jointly optimizing the noisy poses. We'll include SPARF in the Appendix, as you suggested.
>
> However, **the problem setting of SPARF involves per-scene optimization, which is beyond the scope of our approach**. Our method, along with the baseline approaches we compare, assumes a feed-forward solution, making it difficult to directly compare with per-scene optimization approaches. In addition, SPARF considers **noisy** pose as input, which requires a reasonable starting point, while ours assumes **pose-free** scenarios.
>
> ---
>
> **Response to Q4**
>
> Apologies for the confusion. To clarify, the **experimental setting for this study is based on the DTU dataset with large camera baselines.** We didn’t include this in the ablation study, as they are already established technique and commonly used in building cost volumes. However, we plan to add them to the supplementary materials for the final revision, as they provide meaningful improvements.
>
> Also, regarding the values you mentioned, we normalized the camera baseline following the baseline methods (PixelSplat, MVSplat) during the discussion period for the cross-dataset generalizability, which led to slight value changes across all datasets. Specifically, the values you mention has been changed from 19.09dB → 18.78dB (refer to larger baseline results in Table 1). Accordingly, we have to re-run the mean-variance experiments. However, as these results will require 2-3 more days of work, we won’t be able to include them within the remaining discussion period. Nonetheless, we expect similar trends in the updated experiments and will be included in the revised manuscript.

---

### Official Review · Reviewer_rDog · 2024-11-01

**Soundness:** 3
**Presentation:** 3
**Contribution:** 3
**Rating:** 5
**Confidence:** 4

**Summary:**

This paper introduces SHARE, a framework for pose-free generalizable 3D Gaussian Splatting that addresses the challenge of multi-view 3D reconstruction from unposed images. SHARE's key innovation is a ray-guided multi-view fusion network that consolidates multi-view features into a unified pose-aware canonical volume, bridging 3D reconstruction and ray-based pose estimation. It also proposes an anchor-aligned Gaussian prediction strategy for fine-grained geometry estimation within a canonical view. The paper reports that SHARE achieves state-of-the-art performance in pose-free generalizable Gaussian splatting through experiments on diverse real-world datasets, including DTU and RealEstate10K.

**Strengths:**

- The paper presents a novel framework that addresses the challenge of pose-free generalizable 3D Gaussian Splatting, which is an under-explored field in 3D scene reconstruction and novel view synthesis.
- The approach of using a ray-guided multi-view fusion network to consolidate features into a canonical volume for Gaussian prediction is creative.
- The language is clear and technical terms are well-defined, making the paper accessible to readers familiar with the field.

**Weaknesses:**

- Insufficient baselines and experiments.
I think the paper lacks the comparison with the state-of-the-art pose-free multi-view reconstruction framework, i.e., DUST3R [1] (or its subsequent work MAST3R [2]), in terms of pose estimation accuracy and reconstruction quality. Also, several recent works built upon DUSt3R also explored pose-free generalizable Gaussian Splatting, e.g., Splatt3R [3] and InstantSplat [4], I believe that including experimental results and discussions on these methods (at least Splatt3R since it is feed-forward) would make the paper's claim stronger.

- Potentially unfair comparison with pixelSplat and MVSplat.
The authors report the view synthesis results of pixelSplat and MVSplat using "poses predicted by our method" in Table 1 and Table 2. However, we are not clear about the quality of the pose prediction results of SHARE due to the lack of evaluations on pose estimation accuracy. What if we feed (potentially) more robust predicted poses to them, such as the outputs of MAST3R?
Besides, I notice that the results on DTU in Figure 5 are from 3 input views, while the original pixelSplat and MVSplat models were trained on paired images. How did the authors adapt them to 3 input views?

- Small camera baselines and scalability.
The proposed framework utilizes plane-sweep volumes and predicts all Gaussians from a canonical feature volume, raising concerns on its reconstruction capability on more challenging input views, such as large camera baselines and occusions. The qualitative results shown in the paper demonstrate small camera movements compared to the input view, I hope the authors can include some discussions on the upper limit and scalability to more diverse datasets of the proposed method.

[1] Wang, Shuzhe, et al. "Dust3r: Geometric 3d vision made easy." Proceedings of the IEEE/CVF Conference on Computer Vision and Pattern Recognition. 2024.
[2] Leroy, Vincent, Yohann Cabon, and Jérôme Revaud. "Grounding Image Matching in 3D with MASt3R." arXiv preprint arXiv:2406.09756 (2024).
[3] Smart, Brandon, et al. "Splatt3r: Zero-shot gaussian splatting from uncalibarated image pairs." arXiv preprint arXiv:2408.13912 (2024).
[4] Fan, Zhiwen, et al. "Instantsplat: Unbounded sparse-view pose-free gaussian splatting in 40 seconds." arXiv preprint arXiv:2403.20309 (2024).

**Questions:**

My main questions have been listed in the weakness part, I will adjust my final rating accoding to the author's response. I suggest the authors to visualize all the input images in Figure 5 and Figure 2 of the supplementary, intead of labeling "Input view (1/3)" on the top. It is hard for readers to measure the view synthesis quality from only one input view.

---

> ### Author Response · Authors · 2024-11-22
> **Response to Reviewer rDog (1/2)**
>
> > **Insufficient baselines and experiments**
>
> We appreciate the reviewer’s suggestion to include a comparison with additional baselines.
>
> We have compared our work with the concurrent work Splatt3R[1], which utilizes pre-trained MASt3R[2] weights for geometry estimation. We observed that Splatt3R faces a significant scale-ambiguity issue when applied to out-of-distribution data not seen during the training of MASt3R. The estimated scale of the reconstructed point clouds often misaligns with the scale of the camera poses for novel view rendering.
>
> To address this, we attempted to fine-tune Splatt3R on the target datasets (RealEstate10K and DTU) using photometric loss. However, this approach led to convergence issues, with the model output blurry reconstructions. This behavior can be attributed to Splatt3R's reliance on geometry estimation from MASt3R, which requires ground-truth dense depths to mitigate the scale-ambiguity issue. Unfortuantely, our target datasets present challenges in this regard: RealEstate10K lacks ground-truth depths, and DTU provides only sparse, masked depth maps, making it difficult to adapt Splatt3R directly without significant modifications.
>
> - Table A
> | DTU | PSNR ↑ | SSIM ↑ | LPIPS ↓ |
> |---|---|-----|----|
> | Splatt3R | 11.78 | 0.28 | 0.57 |
> | Ours | **17.50** | **0.34** | **0.48** |
> - Table B
> | RealEstate10K | PSNR ↑ | SSIM ↑ | LPIPS ↓ |
> |----|----|----|---|
> | Splatt3R | 15.80 | 0.53 | 0.30 |
> | Ours | **21.23** | **0.71** | **0.26** |
> - Table C
> | **ACID** | **Training Data** | PSNR ↑ | SSIM ↑ | LPIPS ↓ |
> |---|---|---|--|---|
> | Splatt3R | ScanNet++ | 17.49 | 0.63 | **0.26** |
> | Ours | RealEstate10K | **23.47** | **0.69** | **0.26** |
>
> To provide a fair baseline, we evaluated the pre-trained Splatt3R model (trained on ScanNet++) directly on our datasets under its original training conditions. We included both in-dataset (Table A and B) and cross-dataset (Table C) generalization tests. We have included these results in the supplementary material, **Appendix A.4 (Table 7, 8, Figure 12, 13)**, with a detailed discussion of the experimental settings, evaluation metrics, and qualitative visualizations.
>
> Tables A and B show that our method significantly outperforms both datasets by a large margin. In addition, to discuss the cross-dataset generalization quality, we tested our method (trained on RealEstate10K) and Splatt3R (trained on ScanNet) on the ACID[3] dataset. The table shows that our method shows superior performance in all metrics, underscoring the robustness and generalizability of our approach. These results validate our method’s effectiveness in pose-free multi-view reconstruction, even in challenging scenarios without ground-truth depth supervision.
>
> [1] Smart, Brandon, et al. "Splatt3r: Zero-shot gaussian splatting from uncalibrated image pairs." arXiv. 2024.
>
> [2] Leroy, Vincent, Yohann Cabon, and Jérôme Revaud. "Grounding image matching in 3d with mast3r." ECCV, 2024.
>
> [3] Liu, Andrew, et al. "Infinite nature: Perpetual view generation of natural scenes from a single image." *ICCV. 2021.
>
> ---
>
> > **Potentially unfair comparison with pixelSplat and MVSplat**
>
> We understand the reviewer’s concerns and have included the evaluation results of the baselines with the pose inferred from various state-of-the-art pose estimators including DUSt3R [1] and MASt3R [2].
>
> |Method|Pose|Rot. ↓|Trans. ↓|PSNR ↑|SSIM ↑|LPIPS ↓|
> |---|---|---|---|---|---|---|
> | **PixelSplat** | GT | - | - | 20.96 | 0.65 | 0.31 |
> |  | COLMAP | 7.10 | 31.62 | 13.49 | 0.34 | 0.66 |
> |  | MASt3R | 2.40 | **3.52** | 15.69 | 0.40 | 0.50 |
> |  | DUSt3R | **1.77** | 13.66 | 15.98 | 0.42 | 0.47 |
> |  | Ours | 2.74 | 6.28 | 13.29 | 0.31 | 0.66 |
> | **MVSplat** | GT | - | - | 21.00 | 0.69 | 0.24 |
> |  | COLMAP | 7.10 | 31.62 | 14.69 | 0.44 | 0.46 |
> |  | MASt3R | 2.40 | **3.52** | 13.31 | 0.31 | 0.58 |
> |  | DUSt3R | **1.77** | 13.66 | 13.22 | 0.32 | 0.58 |
> |  | Ours | 2.74 | 6.28 | 14.08 | 0.33 | 0.51 |
> | **Ours** |  | 2.74 | 6.28 | **19.94** | **0.63** | **0.28** |
>
> While some pose estimation methods, such as MASt3r, demonstrate higher pose estimation accuracy compared to our approach, the rendering quality using their estimated poses combined with baselines (e.g., PixelSplat, MVSplat) falls significantly short of the results achieved with our method. To ensure a fairer comparison, we have updated the pred-pose baseline in our paper (**Table 1, 2**) to utilize poses from DUSt3R[1], which generally achieve better performance on DTU and re10k datasets.
>
> We have also included a detailed discussion on the implementation of the baselines in **Appendix A.2** of the revised manuscript. We hope this additional clarification addresses your concerns regarding the comparison with the baselines.
>
>
> [1] Wang, Shuzhe, et al. "Dust3r: Geometric 3d vision made easy." CVPR, 2024.
>
> [2] Leroy, Vincent, Yohann Cabon, and Jérôme Revaud. "Grounding image matching in 3d with mast3r." ECCV, 2024.

---

> ### Author Response · Authors · 2024-11-22
> **Response to Reviewer rDog (2/2)**
>
> > **I notice that the results on DTU in Figure 5 are from 3 input views, while the original pixelSplat and MVSplat models were trained on paired images. How did the authors adapt them to 3 input views?**
> >
>
> Since PixelSplat and MVSplat predict depths from each viewpoint and transform to fuse with GT poses, it is naturally adaptable to different number of viewpoints. This is also included in their official Github repository. For our baseline, we trained PixelSplat and MVSplat with 3 views for DTU dataset.
>
> ---
>
> > **The proposed framework utilizes plane-sweep volumes and predicts all Gaussians from a canonical feature volume, raising concerns on its reconstruction capability on more challenging input views, such as large camera baselines and occusions. The qualitative results shown in the paper demonstrate small camera movements compared to the input view.**
> >
>
> While it is generally acknowledged that plane-sweep volume methods can be sensitive to large camera baselines, we would like to highlight that our backbone leverages matching features [1] and a correlation-based cost volume, effectively addressing these challenges as also addressed in prior works (UFORecon [2], CoPONeRF [3]). To validate our method’s adaptability to large camera baseline, we have added the qualitative results on large-baseline inputs in **Figure 14** of **Appendix A.4**.
>
> [1] Xu, Haofei, et al. "Unifying flow, stereo and depth estimation." TPAMI, 2023.
>
> [2] Na, Youngju, et al. "UFORecon: Generalizable Sparse-View Surface Reconstruction from Arbitrary and UnFavOrable Data Sets." CVPR, 2024.
>
> [3] Hong, Sunghwan, et al. "Unifying Correspondence, Pose and NeRF for Pose-Free Novel View Synthesis from Stereo Pairs." CVPR, 2024.
>
> ---
>
> > **I hope the authors can include some discussions on the upper limit and scalability to more diverse datasets of the proposed method.**
> >
>
> As noted in the shared official response, we acknowledge the importance of extending the proposed method to more diverse datasets and scenarios.
>
> Regarding upper limitation, we recognize the challenges in applying our method directly to complex scenarios such as sparse 360-degree input images, as these involve significantly different geometric and visual conditions (e.g., occlusion, extremely low overlap). Recent advances, such as MVSplat360 [1], have shown promise in addressing these scenarios by integrating generative priors for improved 360-degree synthesis. We believe that combining our approach with such methods could offer enhanced performance, particularly for challenging cases involving sparse and wide-baseline inputs.
>
> We appreciate your suggestion and we've included more detailed discussions on these topics in the discussion section of the revised manuscript.
>
> [1] Chen, Yuedong, et al. "MVSplat360: Feed-Forward 360 Scene Synthesis from Sparse Views." NeurIPS, 2024
>
> ---
>
> > **I suggest the authors to visualize all the input images in Figure 5 and Figure 2 of the supplementary, intead of labeling "Input view (1/3)" on the top. It is hard for readers to measure the view synthesis quality from only one input view.**
> >
>
> We thank the reviewer for the thoughtful suggestion. In the **Figure 5** of the main paper, we included up to two input views to prevent the image size from becoming too small to read clearly. Meanwhile, the figures in the **Appendix A.4** include all input views. We hope this revision has improved the overall presentation.

---

> ### Author Response · Authors · 2024-11-26
> **Kind reminder for reviewer-author discussion**
>
> Dear Reviewer rDog,
>
> As the discussion period is ending soon, we wanted to kindly remind you of our responses to your comments. We truly value your feedback and are happy to answer any remaining questions or concerns you might have.
>
> Please feel free to let us know if there is any more information we can provide to help with the discussion.
>
> Thank you again for your time and thoughtful review.
>
> Best regards,
> Authors

---

> ### Author Response · Authors · 2024-11-28
> **Official Comment for Reviewer rDog**
>
> As the PDF revision period is drawing to a close and the discussion deadline is approaching, we would like to kindly remind you of our improvements. We first want to thank your insightful feedback, which has greatly contributed to improving our paper. Based on your suggestions, we have made the following enhancements:
>
> - We have incorporated a recent concurrent work, **Splatt3R**[1], into our baseline comparison. This allows us to demonstrate that our approach outperforms the existing pose-free novel view synthesis method.
>
> - We combined pose-dependent **baselines (pixelSplat[2], MVSplat[3]) with various state-of-the-art pose estimators**, enhancing the thoroughness of the evaluation and baselines. This experiment further supports our claim that simply combining pose estimators with pose-dependent methods leads to geometry misalignment.
>
> - To strengthen the presentation of our results, we included **qualitative results for large baselines**, improving the overall clarity of our paper.
>
> - We conducted additional experiments on datasets such as ACID[4] and BlendedMVS[5], further demonstrating the **cross-dataset generalizability** of our approach.
>
> We are pleased with the improvements we have made to the paper as a result of your valuable feedback. Once again, thank you for your thoughtful comments, and we kindly ask that you consider reflecting on these changes and the resulting experimental improvements in your review scores.
>
> ---
>
> [1] Smart, Brandon, et al. "Splatt3r: Zero-shot gaussian splatting from uncalibrated image pairs." arXiv. 2024.
>
> [2] Charatan, David, et al. "pixelsplat: 3d gaussian splats from image pairs for scalable generalizable 3d reconstruction." CVPR, 2024.
>
> [3] Chen, Yuedong, et al. "Mvsplat: Efficient 3d gaussian splatting from sparse multi-view images." ECCV, 2024.
>
> [4]  Liu, Andrew, et al. "Infinite nature: Perpetual view generation of natural scenes from a single image." ICCV. 2021.
>
> [5]  Yao, Yao, et al. "Blendedmvs: A large-scale dataset for generalized multi-view stereo networks." CVPR. 2020.

---

> ### Comment · Reviewer_rDog · 2024-12-02
> **Response to authors**
>
> Thank you for your detailed responses. However, I must raise a concern regarding the experimental comparison with Splatt3R. Since MASt3R generates point clouds with an arbitrary scale factor, proper evaluation requires aligning ground truth camera poses with the scale of estimated camera poses for novel view rendering. Based on Splatt3R's reported metrics and the visualizations presented in Figures 12 and 13, it appears this camera pose alignment step has been omitted from your evaluation protocol. This oversight could significantly impact the fairness of the comparison, making the current experimental results and your analysis not convincing enough.

---

> > ### Author Response · Authors · 2024-12-03
> > **Official Comment for Reviewer rDog**
> >
> > Thank you for your thoughtful and detailed comments. We appreciate your valuable suggestion for improving the fairness of the evaluation.
> >
> > First, we would like to clarify the issue of scale ambiguity in Splatt3R. They rely on point clouds predicted by the pre-trained MASt3R, which is designed to generate point clouds in a metric scale. However, due to inherent inaccuracies in the prediction process, there is a discrepancy between the estimated scale of the point clouds and the ground-truth scale. This misalignment leads to poor rendering, particularly in the form of distorted or inconsistent results.
> >
> > In fact, to address this, we included a pose rescaling step as we found that directly using the ground-truth pose scale led to render black images. Therefore, during the rendering process, we manually rescale the ground-truth poses by normalization based on the scale of the predicted point clouds.
> >
> > We acknowledge that this scale ambiguity is an inherent limitation of Splatt3R. However, as you pointed out, rescaling the poses based on the predicted scale could offer a more consistent and fairer evaluation. In response, we have conducted additional experiments using rescaled target poses derived from the predicted camera poses. We denote with **Splatt3R*** in below table for the rescaled target pose with the predicted pose.
> >
> > - Table A
> > | DTU | PSNR ↑ | SSIM ↑ | LPIPS ↓ |
> > |---|---|-----|----|
> > | Splatt3R  | 11.78 | 0.28 | 0.57 |
> > | Splatt3R* | 12.53 | 0.38 | 0.49 |
> > | Ours | **17.50** | **0.34** | **0.48** |
> > - Table B
> > | RealEstate10K | PSNR ↑ | SSIM ↑ | LPIPS ↓ |
> > |----|----|----|----|
> > | Splatt3R | 15.80 | 0.53 | 0.30 |
> > | Splatt3R* | 15.14 | 0.48 | 0.39 |
> > | Ours | **21.23** | **0.71** | **0.26** |
> > - Table C
> > | **ACID** | **Training Data** | PSNR ↑ | SSIM ↑ | LPIPS ↓ |
> > |---|---|---|---|---|
> > | Splatt3R | ScanNet++ | 17.49 | 0.63 | **0.26** |
> > | Splatt3R* | ScanNet++ | 19.81 | **0.71** | **0.26** |
> > | Ours | RealEstate10K | **23.47** | 0.69 | **0.26** |
> >
> > Despite these efforts, we found that introducing this modification during evaluation does not fully resolve the scale ambiguity. A key challenge is that predicted poses are not entirely accurate and contain inherent errors. Consequently, an incorrect scaling factor may be derived, leading to errors in the rescaled target pose. These inaccuracies can further introduce rendering issues, as they may distort the relative pose of the target camera, which impacts the final result.
> >
> > In summary, while we agree that the rescaling approach you suggested provides a more consistent evaluation metric, it does not fully eliminate the scale ambiguity inherent in Splatt3R.
> >
> > We hope that our additional experiments and clarifications provide a better justification of the evaluation procedure and address your concerns. Thank you again for your constructive feedback.

---

### Author Response · Authors · 2024-11-22
**Common Comments**

We sincerely appreciate the reviewers for their insightful comments and constructive feedback, which have significantly enhanced the clarity and depth of our work. Below, we address the comments that are highly relevant to all reviewers, while reviewer-specific feedback is addressed individually.

---

### Cross-Dataset Generalization

All reviewers note the lack of the experiments on dataset generalizability, including large-scale or cross-dataset generalization performance. To address this, we conducted cross-dataset experiments, evaluating our model trained with RealEstate10K dataset on ACID[1] dataset, and  our model trained with DTU on BlendedMVS[2] dataset, following established practices in the field.

|  |  | **RealEstate10K → ACID** |  |  | **DTU → BlendedMVS** |  |  |
| --- | --- | --- | --- | --- | --- | --- | --- |
| Method | Pose | **PSNR↑** | **SSIM↑** | **LPIPS↓** | **PSNR↑** | **SSIM↑** | **LPIPS↓** |
| **PixelSplat** | GT | 26.84 | 0.81 | 0.18 | 11.64 | 0.20 | 0.67 |
|  | σ = 0.01 | 21.73 | 0.57 | 0.28 | 11.65 | 0.20 | 0.68 |
| **MVSplat** | GT | 28.18 | 0.84 | 0.15 | 12.04 | 0.19 | 0.56 |
|  | σ = 0.01 | 21.65 | 0.57 | 0.27 | 11.92 | 0.20 | 0.59 |
| **Ours** | - | 23.47 | 0.69 | 0.26 | 12.19 | 0.26 | 0.61 |

As shown in the table, our method exhibits strong generalizability, performing comparably to or even surpassing the baselines that utilize GT poses. We also compared the baseline methods with minimal gaussian noise level (sigma=0.01), where rotation and translation angular errors are far lower than the state-of-the-art pose estimators. We included the comprehensive quantitative (**Table 6**) and qualitative results (**Figure 11**) in the **Appendix A.4.**

---

[1] Liu, Andrew, et al. "Infinite nature: Perpetual view generation of natural scenes from a single image." ICCV. 2021.

[2] Yao, Yao, et al. "Blendedmvs: A large-scale dataset for generalized multi-view stereo networks." CVPR. 2020.

---

### Note · Authors · 2025-02-03

**Comment:**

We would like to withdraw our submission from ICLR OpenReview. Thank you for your consideration.

**Withdrawal Confirmation:**

I have read and agree with the venue's withdrawal policy on behalf of myself and my co-authors.